# Social learning mechanisms shape transmission pathways through replicate local social networks of wild birds

Kristina B Beck*, Ben C Sheldon, Josh A Firth

Edward Grey Institute of Field Ornithology, Department of Biology, University of Oxford, Oxford, United Kingdom

**Abstract** The emergence and spread of novel behaviours via social learning can lead to rapid population-level changes whereby the social connections between individuals shape information flow. However, behaviours can spread via different mechanisms and little is known about how information flow depends on the underlying learning rule individuals employ. Here, comparing four different learning mechanisms, we simulated behavioural spread on replicate empirical social networks of wild great tits and explored the relationship between individual sociality and the order of behavioural acquisition. Our results reveal that, for learning rules dependent on the sum and strength of social connections to informed individuals, social connectivity was related to the order of acquisition, with individuals with increased social connectivity and reduced social clustering adopting new behaviours faster. However, when behavioural adoption depends on the ratio of an individuals' social connections to informed versus uninformed individuals, social connectivity was not related to the order of acquisition. Finally, we show how specific learning mechanisms may limit behavioural spread within networks. These findings have important implications for understanding whether and how behaviours are likely to spread across social systems, the relationship between individuals' sociality and behavioural acquisition, and therefore for the costs and benefits of sociality.

*For correspondence:
kbbeck.mail@gmail.com

Competing interest: The authors declare that no competing interests exist.

## Editor's evaluation

This valuable study will be of interest to researchers in the fields of behavioural ecology, social ecology and evolution, and network science. The authors use simulations on empirically-recorded great tit social networks to examine how behavioural contagion might spread through social groups if individuals follow different social learning rules. The evidence supporting the conclusions is convincing, with careful modeling and parameterization for the chosen system.

## Introduction

Social learning, in which individuals learn from others, is widespread in the animal kingdom, and enables individuals to acquire novel behaviours facilitating phenotypic change (*Heyes, 1994*; *Hoppitt and Laland, 2013*; *Whiten, 2021*). Socially induced changes in behaviour can spread through a population and social networks provide the pathways along which behaviour can spread (*Hasenjager et al., 2021*). Research increasingly shows how the structure of social networks and individual sociality can together influence information flow (*Aplin et al., 2012*; *Evans et al., 2021*; *Kulahci et al., 2016*; *Romano et al., 2018*; *Voelkl and Noë, 2008*). However, information can spread via various social learning mechanisms (*Cantor et al., 2021*; *Evans et al., 2021*; *Firth, 2020*; *Nunn et al., 2009*) and we know surprisingly little about how the relationship between sociality and information flow depends on the social learning mechanisms at play.

By definition, the social spread of behaviour to a focal individual requires contact with at least one knowledgeable individual. Frequently, it is intuitively assumed that the extent (i.e. number and duration) of social contacts to knowledgeable others predicts the likelihood of adoption (*Coussi-Korbel and Fragaszy, 1995*; *Franz and Nunn, 2009*; *Hasenjager et al., 2021*). In this way, behavioural spread is predicted to follow a similar pattern to the transmission of many diseases. However, in contrast to disease spread, individuals may employ 'social learning rules' where they can actively shape how to act on acquired novel information. For example, for more costly behaviours such as the usage of a novel food type, an individual may only change its behaviour after the majority of its social contacts consumes the novel food. Consequently, behavioural spread may require exposure to multiple sources (rather than just one) and depends on the ratio of connections to both informed and uninformed individuals (rather than just the connections to informed others) (*Centola and Macy, 2007*; *Firth et al., 2020*; *Guilbeault et al., 2018*; *Hodas and Lerman, 2014*). Therefore, the type of behaviour considered and its underlying learning rule can fundamentally influence whether and how behaviour spreads through a social network (*Centola and Macy, 2007*; *Firth et al., 2020*).

In sociology, research increasingly demonstrates that the spread of various behaviours, from innovations, to health, and political movements (*Guilbeault et al., 2018*), follow diverse and more complex learning rules compared to the assumptions of many disease models (*Centola and Macy, 2007*). In contrast, research in animal systems has rarely explored how the diffusion dynamics of behaviours may be altered by learning rules (but see *Nunn et al., 2009*; *Cantor et al., 2021*; *Evans et al., 2020*; *Evans et al., 2021*), which is somewhat surprising given that previous studies have revealed several social learning strategies in animals that suggest a range of different underlying social learning mechanisms (*Hoppitt and Laland, 2013*; *Kendal et al., 2018*). For instance, an increasingly reported learning mechanism is conformist learning in which individuals disproportionally adopt the behaviour performed by the majority of their social connections (e.g. sticklebacks: *Pike and Laland, 2010*; chimpanzees: *Haun et al., 2012*; vervet monkeys: *van de Waal et al., 2013*; great tits: *Aplin et al., 2015a*; fruit flies: *Danchin et al., 2018*). Further, individuals often only learn from specific individuals (e.g. depending on status, *Canteloup et al., 2020*; relatedness, *Wild et al., 2019*; or conspecifics, *Farine et al., 2015*) or adopt behaviours only once the social connections to informed others surpass a certain threshold (*Rosenthal et al., 2015*).

Research on social learning in animal social networks has frequently assumed that more social individuals (i.e. with more social connections and central network positions) have a higher probability to adopt new behaviours because they are more likely to hold connections to knowledgeable others compared to less social individuals (*Aplin et al., 2012*; *Claidière et al., 2013*; *Kulahci and Quinn, 2019*). This link between individual sociality and behavioural adoption can be expected if the learning rule depends on the sum and strength of social connections to knowledgeable others. However, this relationship may change when learning rules rely on both the connections to informed and uninformed individuals (*Centola and Macy, 2007*; *Firth, 2020*). For instance, in the case of conformist learning, we may expect that the most social individuals will be less likely to adopt (because it may take longer until the majority of their social connections becomes informed). Such patterns have been reported in humans, where highly connected individuals required stronger social signals in order to act on information (*Hodas and Lerman, 2014*; *Hodas and Lerman, 2012*) and poorly connected individuals may utilize information sooner (*González-Avella et al., 2011*). Hence, predictions of how individual sociality relates to the probability of acquiring novel behaviour, and the resulting transmission pathways, can change fundamentally depending on the social learning mechanism at play (*Centola and Macy, 2007*; *Firth et al., 2020*).

Examining and comparing the transmission pathways of behaviours that follow different learning mechanisms in wild animals is challenging. Therefore, research investigating the relationship between social structure and information flow often simulates behavioural spread (*Cantor et al., 2021*; *Evans et al., 2021*; *Evans et al., 2020*; *Nunn et al., 2009*; *Voelkl and Noë, 2008*). For instance, studies compared the transmission speed (number or proportion of individuals informed at a given timestep) of simple versus conformity learning (*Evans et al., 2021*; *Evans et al., 2020*) or 'prestige' (subordinates copy dominants) versus conformity learning (*Nunn et al., 2009*). While these studies show that on the population level, different learning mechanisms, together with the social network structure, can fundamentally impact the diffusion dynamics (e.g. how quickly a behaviour can spread), we know little

on how learning mechanisms impact the relationship between individual sociality and the probability of behavioural adoption.

In addition, behavioural simulations are often performed on artificial social networks with pre-defined structure and size (*Voelkl and Noë, 2008*; *Nunn et al., 2009*; *Cantor et al., 2021*; *Evans et al., 2021*), and may thus represent unrealistic social structures, failing to capture the social behaviour observed in real animal social networks (but see *Naug, 2008*; *Romano et al., 2018*). Therefore, in addition to purely computational studies, it is important to examine real-world social networks to test whether general findings from artificial networks can be replicated using real social systems. Further, simulated social networks are often relatively large including 100 or more individuals, and empirical social networks may be generated over prolonged periods of time. However, for behavioural spread, an individual's social connections at a relatively small temporal scale may predict subsequent transmission (*Aplin et al., 2015a*; *Aplin et al., 2015b*; *Somveille et al., 2018*). Many animal species live in non-stable social groups such as fission–fusion societies (e.g. various species of birds: *Silk et al., 2014*, primates: *Amici et al., 2008*, and fish: *Papastamatiou et al., 2020*; *Wilson et al., 2014*) where group composition and size frequently change. As a result, social connections between individuals can change over time, and empirical networks generated over weeks/months, and artificial, large networks, may overestimate the social connections of an individual at the time a new behaviour emerges. Thus, it is crucial to examine social networks – both empirical and artificially derived – on a meaningful temporal scale (which will be study species dependent) to better understand whether and how different types of behaviours spread through social networks.

In this study, we explore by simulation how novel behaviours, transmitted according to different social learning mechanisms, spread through replicated empirical social networks of great tits (*Parus major*). Great tits are small songbirds that forage in fission–fusion mixed-species flocks during winter (*Ekman, 1989*) and frequently use social information (e.g. to find novel food: *Aplin et al., 2012*; *Firth et al., 2016*, to access novel food: *Aplin et al., 2015a*, and for prey avoidance: *Hämäläinen et al., 2020*; *Thorogood et al., 2018*) which makes them an ideal study species. Here, we create social networks from empirical data on birds' foraging associations at distinct locations sampled on two days each week to capture the social structure at a relatively small spatiotemporal scale. Subsequently, we simulate behavioural spread on these weekly, local, networks using four different social learning mechanisms and compare how the social behaviour of individual great tits relates to the order in which they acquire novel behaviour under the four different mechanisms.

The first learning mechanism follows the omnipresent concept of simple contagion, which is mainly inspired by models on disease spread and was first formulated in the field of sociology (*Guilbeault et al., 2018*). Simple contagion assumes that the probability of adopting a novel behaviour depends on the number and strength of connections to informed individuals (thereafter simple rule, *Coussi-Korbel and Fragaszy, 1995*; *Franz and Nunn, 2009*; *Hasenjager et al., 2021*). The other three learning mechanisms imply more complex adoption rules (*Centola and Macy, 2007*) where behavioural adoption requires more social reinforcement: (1) a threshold rule, (2) a proportion rule, and (3) a conformity adoption rule. Here, the probability of adopting the novel behaviour depends on: (1) the connections to informed individuals surpassing a given threshold; (2) the proportion of connections to informed individuals (rather than the sum); and (3), the behaviour that the majority of connections performs. Threshold-based learning rules have been studied frequently in sociology and network sciences (*González-Avella et al., 2011*; *Granovetter, 1978*; *Watts, 2002*), but have rarely been considered in animals (*Rosenthal et al., 2015*). In contrast, conformity learning, where individuals are disproportionally more likely to copy the behaviour performed by the majority, has received much attention both in humans (*Boyd and Richerson, 1988*; *Haun et al., 2012*; *Toyokawa and Gaissmaier, 2022*) and animals (*Aplin et al., 2015a*; *Danchin et al., 2018*; *van de Waal et al., 2013*). The proportion rule assumes that the transmission rate is proportional to the ratio of informed and uninformed individuals (rather than disproportional as in the conformity rule) and has rarely been considered (*Centola, 2018*; *Firth, 2020*; *Rosenthal et al., 2015*).

Individual variation in sociality – the number and strength of social connections and centrality within the network – may influence the access to information and thus behavioural adoption. We infer individuals' sociality by extracting three commonly used weighted social network metrics: the weighted degree (i.e. sum and strength of their social connections to others), weighted clustering coefficient (propensity for their associates to be associated with one-another), and weighted betweenness

(propensity to act as a 'bridge' within the network). We predicted that the relationship between individual sociality and behavioural adoption would differ depending on the social learning mechanism. Specifically, if the likelihood to adopt a behaviour depends on the number and strength of connections to informed individuals such as in the case for the simple and threshold rule, we predicted that individuals with high degree and betweenness and low clustering coefficient should be faster in adopting the novel behaviour due to being more likely to be connected to at least one informed conspecific. In contrast, if the likelihood of adopting a behaviour depends on the ratio of an individual's informed and uninformed connections, such as in the case for the proportion or conformity rule, we expected that individuals with low degree and betweenness and high clustering coefficient should be faster in adopting the novel behaviour because the majority of their social connections should become informed faster.

## Materials and methods
### Study system
The empirical data used in this study were collected over 3 years (December 2011–March 2014) in a population of great tits located in Wytham Woods, Oxfordshire, UK (51°46' N, 01°20' W, approx.

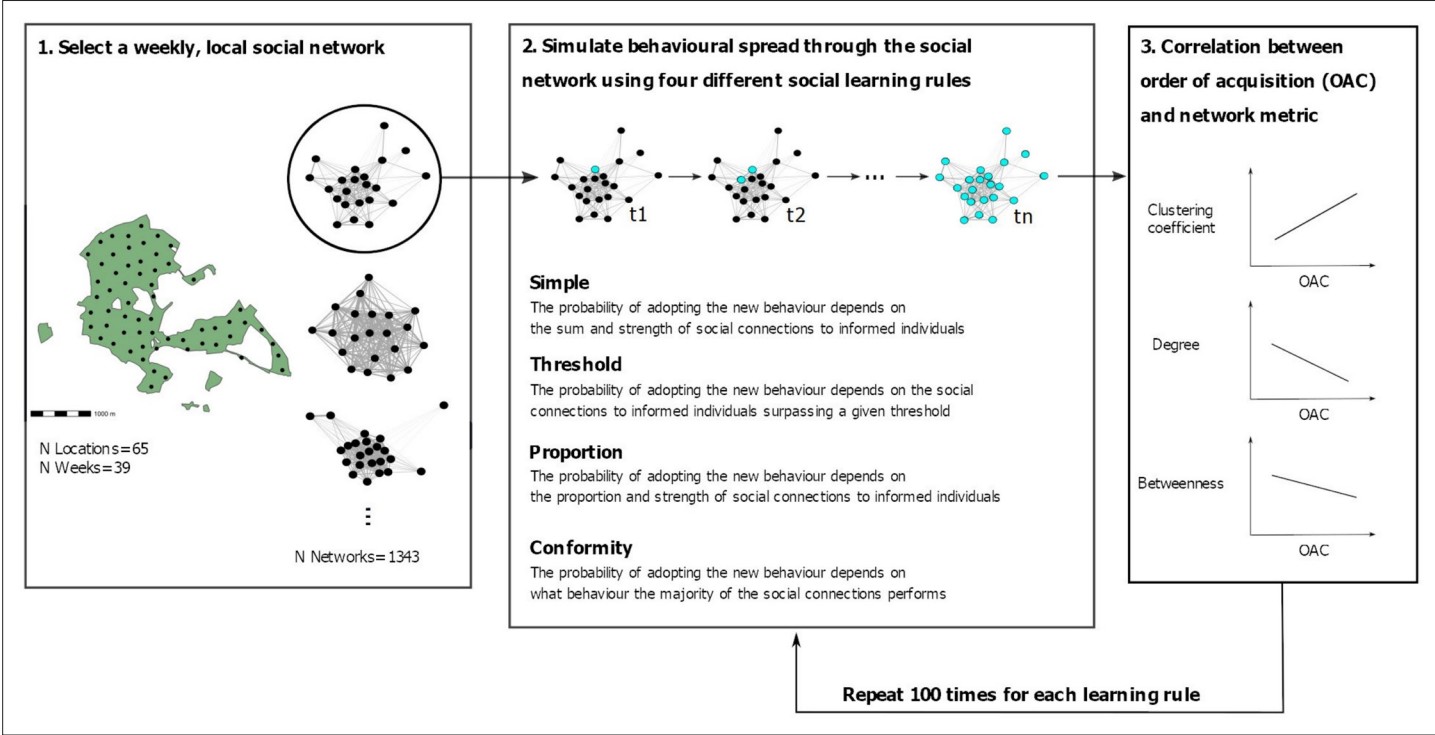

**Figure 1.** Schematic overview of the simulation procedure. First, a weekly social network of one of the feeder locations (shown as black dots) in the study site was selected. Second, behavioural spread was simulated on the selected network using four different social learning rules (i.e. simple, threshold, proportion, and conformity). The starting point (i.e. the first individual performing the new behaviour) was randomly chosen. Then, at each timestep (t1–tn), a naive individual adopted the novel behaviour with a given probability of the adoption event being from social learning (dependent on the social learning rule at play; see methods for further details) until all individuals in the network had adopted the novel behaviour. Third, we calculated a correlation coefficient (Spearman's rank correlation coefficient) between three individual social network metrics (i.e. weighted clustering coefficient, weighted degree, and weighted betweenness) and the order (i.e. timestep) in which individuals adopted the novel behaviour. Finally, we repeated this process 100 times for each weekly, local social network.

The online version of this article includes the following figure supplement(s) for figure 1:

**Figure supplement 1.** Great tit social networks illustrating individuals' different social network positions.

**Figure supplement 2.** Data distribution of four global network metrics from the 1343 weekly, local social networks.

**Figure supplement 3.** Mean correlation coefficient between weighted degree and order of acquisition in relation to different numbers of simulations.

**Figure supplement 4.** Data distribution of network sizes (i.e. the number of individuals each network contained).

385 ha). Great tits are short-lived (mean lifespan of 1.9 years, *Bulmer and Perrins, 1973*) hole-nesting songbirds that form socially monogamous pairs, and establish territories during the breeding season (March–June). During the non-breeding season (September–February), great tits forage with other species in loose fission–fusion flocks that differ in size and composition (*Ekman, 1989*; *Hinde, 1952*) and consist of mainly unrelated individuals (annual population turnover of about 50% and less than 1.5% of social foraging associations are between first-order relatives, *Firth and Sheldon, 2016*). Great tits frequently use social information in foraging contexts (*Aplin et al., 2012*; *Farine et al., 2015*; *Firth et al., 2016*; *Thorogood et al., 2018*).

The woodland contains 1017 nest boxes hosting breeding great tits and 65 bird feeders that were deployed during the winter months in an evenly spaced grid (see *Figure 1*). Each feeder contained two access holes of which both were equipped with radio-frequency identification (RFID) antennas. The feeders were in place from December to February across three winters (2011–2012, 2012–2013, and 2013–2014) and collected data on the bird visits 2 days each week (from pre-dawn Saturday morning until after dusk on Sunday evening) resulting in 13 sampling periods each year. At other times feeders were closed. For the duration of the study, the location of each feeder was consistent.

All birds were caught in either a nest-box or a mist-net and were fitted with a uniquely numbered metal leg ring (British Trust for Ornithology). In addition, each bird was also fitted with a uniquely coded passive integrated transponder (PIT) tag enclosed in a plastic ring fitted to the other leg. This allowed us to record each visit of a PIT-tagged bird when it came close to the RFID antenna of a feeder (approximately 3 cm). At every detection, the bird's unique PIT tag code, and the date and time were saved to a data logger. Breeding surveys and frequent trapping allowed to fit almost all individuals with metal rings and PIT tags (>90%, *Aplin et al., 2013a*).

## Social networks

We created social networks based on the foraging associations of PIT-tagged great tits at each feeder and each of the 13 weekends across the 3 years. We created temporal and locally restricted networks because we wanted to generate a large number of different social networks (rather than just one network from the whole population) and because we expect networks from such a small time-window (i.e. one weekend) to be most meaningful in capturing the social connections relevant for the spread of a novel behaviour. For instance, *Aplin et al., 2015b* showed that individuals disproportionally copied the behaviour of the majority of individuals in the social group that preceded the focal individual's first successful solve. Across the winter, individual great tits may move between locations and new individuals arrive at different times to the study site. Therefore, generating a social network spanning the whole study period will contain many connections not relevant at the time a novel behaviour emerges. Further, when examining the relationship between individual sociality and the probability of adoption, generating a social network from the whole population would add considerable spatial noise. For instance, within a sub-population where a new behaviour emerges, individuals with high connectivity may be faster in adopting the behaviour. However, if examined on the population level, such a relationship may be obscured by spatial effects, because an individual's probability of behavioural adoption will be considerably predicted by its' spatial proximity to the location of behavioural emergence. Further, creating social networks for each feeder location provided a comparable spatial unit and did not require to draw arbitrary spatial boundaries across the study site.

All analyses were conducted in R 4.0.5 (*R Development Core Team, 2020*). An 'association' was defined as two birds foraging together within the same flock. Flock membership was identified using Gaussian Mixture Models (*Psorakis et al., 2015*; *Psorakis et al., 2012*) from the R package 'asnipe' (*Farine, 2013*). This method detects events of increased feeding activity in the spatiotemporal data, clusters these into non-overlapping gathering events (i.e. flocking events), and assigns each individual detection to the event it most likely belonged to. This provided us with information about which individuals co-occurred in the same flock (*Psorakis et al., 2015*; *Psorakis et al., 2012*). From the pattern of co-occurrences, we then inferred the strength of associations for each dyad. We calculated association strength using the simple ratio index (SRI, *Cairns and Schwager, 1987*). The SRI describes the proportion of observations of two individuals in which they were seen together, ranging from 0 (never observed in the same flock) to 1 (always observed in the same flock). We inferred a proportional measure for the association strength between individuals (rather than just a measure for the total number of times two individual were observed together) because we have an unequal number of

observations for each individual (***Farine and Whitehead, 2015***; ***Hoppitt and Farine, 2018***). Further, the SRI provides a more representative measure for the social relationship between two individuals across multiple contexts (e.g. while not foraging at the feeder). We created undirected social networks with edges weighted by the SRI for each sampling weekend (in total: 39 weekends across 3 years) and feeder location (in total: 65).

For each of the weekly, local networks, we then inferred for each individual three social network metrics: the weighted clustering coefficient, the weighted degree and the weighted betweenness. All network metrics were calculated using the R package 'igraph' (***Csardi and Nepusz, 2006***). The weighted clustering coefficient was calculated following ***Barrat et al., 2004***. It represents the proportion of the sum of edge weights of all direct connections of a focal individual $i$ over the sum of weights of all connections of individual $i$ that form a triangle (i.e. where two direct connections of individual $i$ are themselves connected). The weighted degree describes the total interaction rate for a focal individual $i$ with all other individuals, defined as the sum of all the focal individual's edge weights. The weighted betweenness describes the number of weighted, shortest paths from all individuals to all other individuals that pass through the focal individual $i$ and measures an individuals' propensity to move between groups. Here, weights were added by considering the inverse of an individuals' edge weights.

Finally, we standardized the individual metrics within each network to allow comparisons between networks. Social networks including fewer than ten individuals and exhibiting no variation in individual network metrics were excluded from further analyses resulting in the final sample size of 1343 social networks (generated from 62 locations). The three individual network metrics are moderately correlated, with weighted clustering coefficient being negatively correlated to weighted degree and weighted betweenness, weighted degree and weighted betweenness were positively correlated (see ***Supplementary file 1a***). In addition, we provide example networks with individual great tits colour-coded based on their different weighted network metrics (***Figure 1—figure supplement 1***) and calculated four global network measures (network density, average path length, average edge weight, and modularity) to provide a general overview of the weekly, local great tit social structures (***Figure 1—figure supplement 2***, details on how these metrics were calculated can be found in the figure legend).

## Simulations

On each network, we simulated behavioural spread using four different social learning rules, as described in ***Firth et al., 2020***, using the R package 'complexNBDA'. A brief explanation is provided below but full detailed description and tests of each can be found in ***Firth et al., 2020*** and all resources are freely available at https://github.com/whoppitt/complexNBDA (***Hoppitt, 2020***). The R code and data to replicate the simulations used in this manuscript can be found at https://osf.io/6jrhz/.

(1) Simple rule: This transmission rule follows the logic of the classic NBDA framework:

$$\lambda_i\left(t\right) = \lambda_o\left(t\right)\left(s\sum_{j=1}^{N}a_{ij}z_j\left(t\right) + 1\right)\left(1 - z_i\left(t\right)\right)$$

Here, $\lambda_i\left(t\right)$ represents the rate at which individual $i$ acquires a novel behaviour as a function of time. $\lambda_o\left(t\right)$ represents a baseline rate function (i.e. the rate of asocial learning at time $t$) and $s$ determines the strength of social transmission. When simulating the order of acquisitions across individuals (OADA) for a specified parameter set instead of the times of acquisitions (TADA) (***Hasenjager et al., 2021***), the probabilities that each specific individual is next to learn is independent of $\lambda_o\left(t\right)$ and thus $\lambda_o\left(t\right)$ drops out of the equation. $z_i\left(t\right)$ is the 'status' of individual $i$ at time $t$, (1 = informed; 0 = naive), and $N$ is the number of individuals in the network. The rate at which an individual acquires a novel behaviour through social learning is proportional to $\sum_{j=1}^{N}a_{ij}z_j\left(t\right)$, the total connections to informed individuals at time $t$. Therefore, $s$ gives the rate of transmission per unit connection relative to the rate of asocial learning of the novel behaviour. For example, when $s = 2$, an increase of 1 in an individuals' edge weights to informed individuals will increase the rate of social learning by 2 times the baseline rate. $\left(1 - z_i\left(t\right)\right)$ ensures that only naive individuals acquire the behaviour. Consequently, the more and stronger connections to informed individuals, the more likely an individual is to adopt the behaviour. The social transmission strength $s$ was set to 5.

Following, we define three more complex rules that generalize the classic NBDA model as:

$$\lambda_i(t) = \lambda_o(t)\left(T\left(a_i, z(t)\right) + 1\right)\left(1 - z_i(t)\right)$$

Here, $a_i$ represents the connections individual $i$ has to all others in the network, $z(t)$ gives the status of each individual in the network at time $t$, and $T\left(a_i, z(t)\right)$ is a transmission function determining how the rate of transmission is determined by $a_i$ and $z(t)$.

(2) Threshold rule: This transmission rule is defined as:

$$T\left(a_i, z(t)\right) = \left(\frac{c}{1 - \frac{1}{1+exp(ab)}}\right)\left(\frac{1}{1 + exp\left(-b\left(\sum_j a_{ij}z_j(t) - a\right)\right)} - \frac{1}{1 + exp(ab)}\right)$$

Similar to the classic NBDA, the rate of social transmission is zero when the total connections to informed individuals, $\sum_j a_{ij}z(t) = 0$. However, the rate of transmission increases suddenly as the threshold, $a$, is approached, to a maximum value of $c$. The parameter $b$ determines how sharp the threshold effect is. Our threshold rule differs from how threshold rules are sometimes defined in network sciences where the threshold represents a true step function rather than a sigmoidal curve. Here, we aimed to generate a model with a clear sharp threshold, so we set $b = 3$ for our simulations (for details and other parameter settings for $b$, see *Firth et al., 2020*). We set the threshold value $a$ to 5 for all networks. This means that an individual's weighted connections to informed individuals need to be ≥5 before an individual's behavioural adoption is likely to stem from social learning. For example, if an individual with two connections with weights of 2 each to informed others adopts the new behaviour, the probability of this adoption event stemming from social learning is low. In contrast, if the individual's two connections have a weight of 3 each, the behavioural adoption likely stemmed from social learning under the set threshold rule (see also *Firth et al., 2020* for more details). The social transmission strength $s$ was set to 5.

(3) Proportion rule: This transmission rule is defined as:

$$T\left(a_i, z(t)\right) = s\frac{\sum_j a_{ij}z_j(t)}{\sum_j a_{ij}}$$

Here, the learning rate is proportional to the ratio of connections that an individual $i$ holds to informed others. As such, the individual with the highest proportion is most likely to learn and assumes additional influence from individuals' uninformed connections rather than just considering the sum of connections to informed individuals such as in the simple and threshold model. The social transmission strength $s$ was set to 5.

(4) Conformity rule: Finally, the fourth transmission rule assumes that individuals are disproportionately more likely to copy the majority of the population (i.e. frequency dependent):

$$T\left(a_i, z(t)\right) = s\frac{\left(\sum_j a_{ij}z_j(t)\right)^f}{\left(\sum_j a_{ij}z_j(t)\right)^f + \left(\sum_j a_{ij}\left(1 - z_j(t)\right)\right)^f}$$

Here, the frequency dependence parameter is $f \geq 1$, and $s > 0$. When $f = 1$ this model reduces to the proportional model above, and as $f$ increases the strength of conformity bias increases. Thus, an individual is expected to adopt a new behaviour if it is perceived as being performed by the majority of its' social connections. Similar to the proportional rule, the conformity rule considers an individual's informed and uninformed connections. Further, in this way, this conformity rule is somewhat analogous to a threshold rule but based on the proportion of informed connections rather than the total connectivity to informed individuals. For our simulations, we set $f$ to 5 and the social transmission strength $s$ to 5.

For each simulation the individual 'initiating' the behaviour (i.e. the demonstrator) was randomly chosen, and was then used across the four transmission models. We then simulated behavioural spread across the entire network under each transmission model separately. This means that at each timestep one new individual adopted the seeded behaviour, whereby each time each individual has a given probability of adopting the behaviour through social learning which ranges from 0 (asocial

acquisition, i.e. an individual has no connections to informed individuals and cannot socially learn the behaviour under the set learning rule) to 1 (social acquisition, that is when an individual has the maximum probability of learning socially under the set learning rule). Therefore, at each timestep, the new individual adopting the behaviour was stochastically chosen based on their probability of adopting the behaviour in the previous timestep (i.e. where the one most likely to be chosen was the one most likely to adopt the behaviour under the given social learning rule). For instance, under the proportion rule, the next individual adopting the behaviour (i.e. from t1 → t2, see *Figure 1*) would most likely be the one with the highest proportion of connections to informed others. Following, we inferred the probability of this adoption event stemming from social learning. For instance, for an individual that adopted the novel behaviour with a proportion of connections to informed others of 0.2, the probability of the behavioural adoption stemming from social learning would be lower compared to the individual having a proportion of 0.7.

Once individuals adopted the novel behaviour, they remained 'informed' within each simulation run. For each network, we repeated the simulations 100 times to minimize the influence of the identity of the randomly selected demonstrator on the subsequent transmission pathways. We selected 100 simulation runs because this was enough in acquiring a relatively stable mean correlation coefficient between network metric and order of acquisition (*Figure 1—figure supplement 3*). We repeated our simulations testing various parameter combinations for $s$ (1, 5, 10), $f$ (3, 5, 7), and $a$ (3, 5, 7) which we consider appropriate for our study system (i.e. the average strength of an individuals' connections is approximately 2.5). In the main text we present results for $s$ = 5, $f$ = 5, $a$ = 5 and as such a total of 537,200 simulations (1343 weekly, local networks × 4 different learning rules × 100 simulation runs). Results for all other parameters can be found in the supplementary material and with the code provided other parameter combinations can be tested.

## Data summary statistics

To assess the relationship between each individual's network metric and the order of acquisition, we calculated Spearman's rank correlation coefficients. After every simulation, for each of the networks and transmission models, we calculated the Spearman's rank correlation coefficient between the order in which individuals adopted the behaviour (always excluding the demonstrator) and each of the three individual network metrics (i.e. the weighted clustering coefficient, the weighted degree and the weighted betweenness). For an overview of the transmission process, see *Figure 1*. For each network and model, we then calculated the average correlation coefficient for each network metric across the 100 simulations (*Figure 1*). To additionally assess the general relationships between the size of the network, and the correlation coefficient between individuals' centralities and their acquisition order, we used linear mixed-effect models using the 'lme4' package (*Bates et al., 2015*). For each model separately, we set the average correlation coefficient as the dependent variable and network size as the predictor variable. Location identity and week nested in year were set as random effects to factor in these differences when assessing this relationship (*Supplementary file 1b*). We examined model assumptions and fit using graphical methods (e.g. *qq* plot of residuals, fitted values versus residual plots, *Korner-Nievergelt et al., 2015*).

## Results

The data in this study were simulated across 1343 empirically derived social networks, inferred from recordings of 1774 individual great tits at 62 feeder locations across 39 weekends and 3 years. Social networks varied in size (right skewed distribution towards smaller networks, see *Figure 1—figure supplement 4*) and consisted of an average of 21.7 individuals (min = 10, max = 77, sd = 10.0; note that we excluded social networks smaller than 10 from the analysis [see methods] resulting in the minimum network size of 10). For each location, we included on average 21.7 networks into the analysis (min = 1, max = 39, sd = 11.9) and each individual was part of on average 16.5 networks (min = 1, max = 88, sd = 13.2). Individuals on average visited 1.3 different feeder locations on a weekend (min = 1, max = 10, sd = 0.7) and from 21,036 occasions where individuals were recorded on a given weekend, individuals had visited only one location in 14,888 occasions (71%).

# Relationship between individual social behaviour and order of acquisition

Simulating behavioural spread under four different social learning rules revealed different transmission pathways across social networks. For both the simple and the threshold rule, the weighted clustering coefficient was on average positively related to the order of acquisition, with more clustered individuals adopting the seeded behaviour later than less clustered individuals across different network sizes (*Figure 2*). Weighted degree and betweenness were on average negatively related to the order of acquisition. Thus, individuals with higher weighted degree and weighted betweenness adopted the novel behaviour on average faster than individuals with a lower weighted degree and betweenness (*Figure 2*). In addition, we show the relationship between mean individual network metrics and the standardized order of acquisition across network sizes (*Figure 2—figure supplement 1*). For the simple learning rule, the relationship between network metric and order of acquisition only changed when the majority of individuals in a network had already adopted the behaviour (with approximately 75% of individuals knowledgeable; *Figure 2—figure supplement 1*). Behavioural spread under the threshold rule showed a small 'hump' shortly after the start of the spread, especially for the network metric weighted degree (*Figure 2*, *Figure 2—figure supplement 1*). Here, the initial individuals to acquire the behaviour exhibited network metrics close to the mean, suggesting that acquisition at the initial stages (when the starting individual is chosen randomly) likely depends on the connections to the demonstrator and/or asocial learning (particularly as social learning may not be likely yet under the set threshold, see section on '*Probability of social spread*'). As more individuals became informed, social learning becomes much more likely as the threshold is possible to be reached, particularly for individuals with higher weighted degree, and possibly higher betweenness and lower clustering, adopted the behaviour sooner (e.g. start of the hump). Finally, the relationship between mean network metric and order of acquisition reversed, suggesting that individuals with higher weighted clustering and lower weighted degree and betweenness adopted the behaviour last. This may partly be a product of necessity (given the opposite type of individuals are already informed). But, interestingly the presence of the 'hump' was most prominent in larger networks and at low thresholds ($a = 3$, *Figure 2—figure supplement 5*), suggesting that under these scenarios this may be related to a larger variation in network positions (or more extremely central individuals) or more opportunities for social learning being present earlier on in the total diffusion (see also section on '*Probability of social spread*'). In contrast to the simple and threshold rules, there was little or no relationship between individual social network metrics and the order of acquisition under both the proportion and conformity learning rules (*Figure 2*, *Figure 2—figure supplement 1*).

Assessing the relationship between each individuals' network metric and the order of acquisition across all social networks and simulation runs, revealed substantial variation in relationship strength (*Figure 3*). Across network metrics, there were on average the strongest positive (*Figure 3*: weighted clustering coefficient) and negative (*Figure 3*: weighted degree and betweenness) correlations under the simple rule, and coefficients for the threshold, proportion, and conformity model were lower. Even though *Figure 2* indicates a clear relationship between average network metric and order of acquisition under the threshold model (*Figure 2*), correlation coefficients were very small (*Figure 3*). This may be because of the non-linear relationship under the threshold rule in which the slope of the relationship between network position and order of acquisition changes direction as more of the population becomes informed (*Figure 2*, *Figure 2—figure supplement 1*), leading to overall low correlation coefficients (*Figure 3*).

The direction of the relationship between average network metrics and order of acquisition remained unchanged when setting lower or higher parameters for the social transmission strength '*s*' (*Figure 2—figure supplements 2 and 3*). However, for the simple rule, the correlation coefficients became on average stronger under larger social transmission rates (*Figure 3—figure supplement 1*). Further, different values for the frequency dependence '*f*' under the conformity rule, and different values for the threshold location '*a*' under the threshold rule did not change the general direction of the relationship between average network metrics and order of acquisition (*Figure 2—figure supplements 4 and 5*, *Figure 3—figure supplement 2*). However, under the threshold model, increasing the threshold location on average reduced the correlation between network metrics and order of acquisition (*Figure 2—figure supplement 5*, *Figure 3—figure supplement 2*).

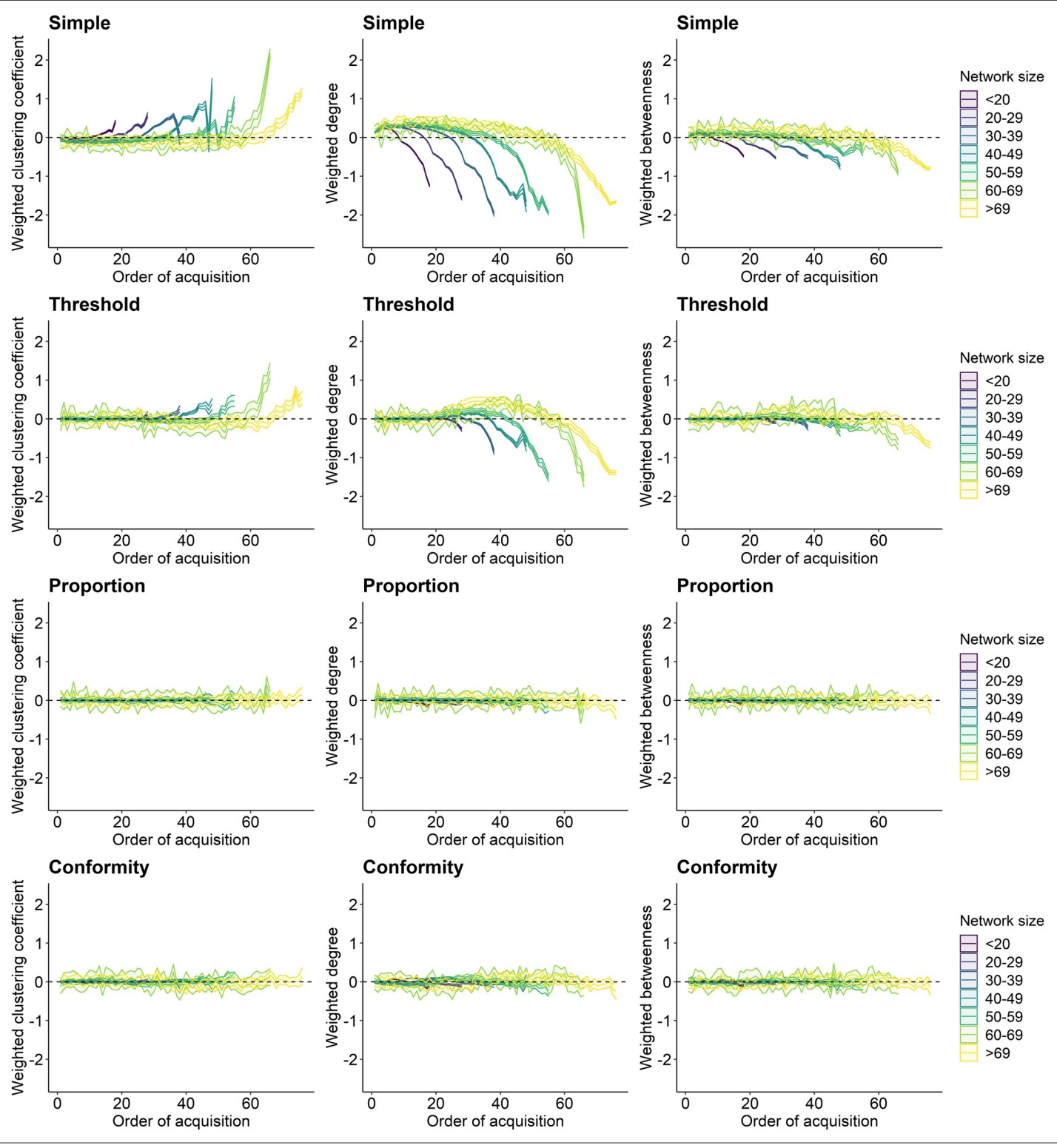

**Figure 2.** Relationship between individual network metric and the order of acquisition for each social learning rule. Each column shows a different network metric (left to right: weighted clustering coefficient, weighted degree, and weighted betweenness). Each row represents one of the four spreading rules (top to bottom: simple, threshold, proportion, and conformity). Lines plot the average network metric for each order of acquisition and ribbons show the 95% confidence interval from the 100 simulations for each binned group of network sizes. Colour represents network size with darker colour indicating smaller networks. The social transmission rate, the threshold location, and the frequency dependence parameter were set to 5.

The online version of this article includes the following figure supplement(s) for figure 2:

*Figure 2 continued on next page*

*Figure 2 continued*

## Relationship between social network size and pathways of behavioural diffusion

The direction and magnitude of the correlation between individual sociality and their order of acquisition were partly predicted by network size (*Figure 4*, *Supplementary file 1b*). For the simple and threshold model, behavioural spread on larger networks led to more positive correlations between individual network metric and order of acquisition for weighted clustering coefficient and more negative correlations for weighted degree and betweenness (*Figure 4*, *Supplementary file 1b*). The predicted effects of network size on mean correlation coefficient inferred under the proportion and conformity rule suggest contrasting directions or no relationship with network size (*Figure 4*, *Supplementary file 1b*).

Overall, the predicted effects of network size on the inferred correlation coefficients were small, particularly for more complex contagions (*Figure 4*, *Supplementary file 1b*). The relationship between correlation coefficient and network size was modulated by the social transmission rate ('*s*' parameter; *Figure 4—figure supplement 1*). For the simple model, the direction of the relationship did not change across different social transmission rates, that is correlation coefficients became on average more positive (weighted clustering coefficient) and negative (weighted degree and weighted betweenness) with increasing network size (*Figure 4—figure supplement 1*). Further, the slope of the relationship between network size and correlation coefficient for each network metric remained relatively constant across social transmission rates (*Figure 4—figure supplement 1*). However, for weighted betweenness, there was no relationship between correlation coefficients and network size

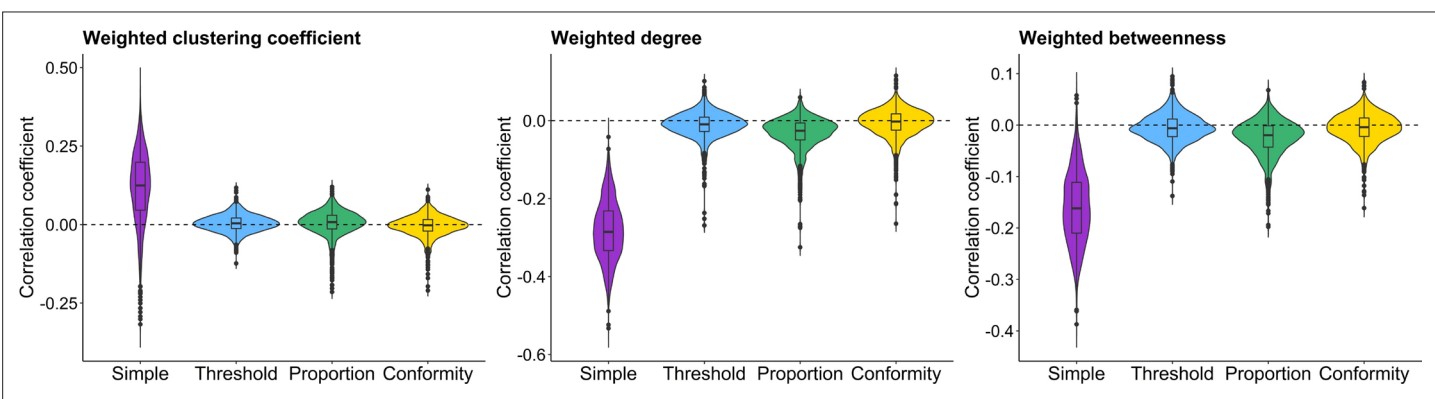

**Figure 3.** Distribution of average correlation coefficients for each social learning rule and network metric. Violin and boxplots show the distribution of the average correlation coefficients between individual network metric and order of acquisition across 100 simulations from each network for each of the four social learning rules (i.e. simple, proportion, conformity, and threshold). Each plot shows one of the individual network metrics (weighted clustering coefficient, weighted degree, and weighted betweenness).

The online version of this article includes the following figure supplement(s) for figure 3:

**Figure supplement 1.** Distribution of average correlation coefficients for each social learning rule and network metric under different social learning rates.

**Figure supplement 2.** Distribution of average correlation coefficients for the threshold and conformity learning rule under different parameters.

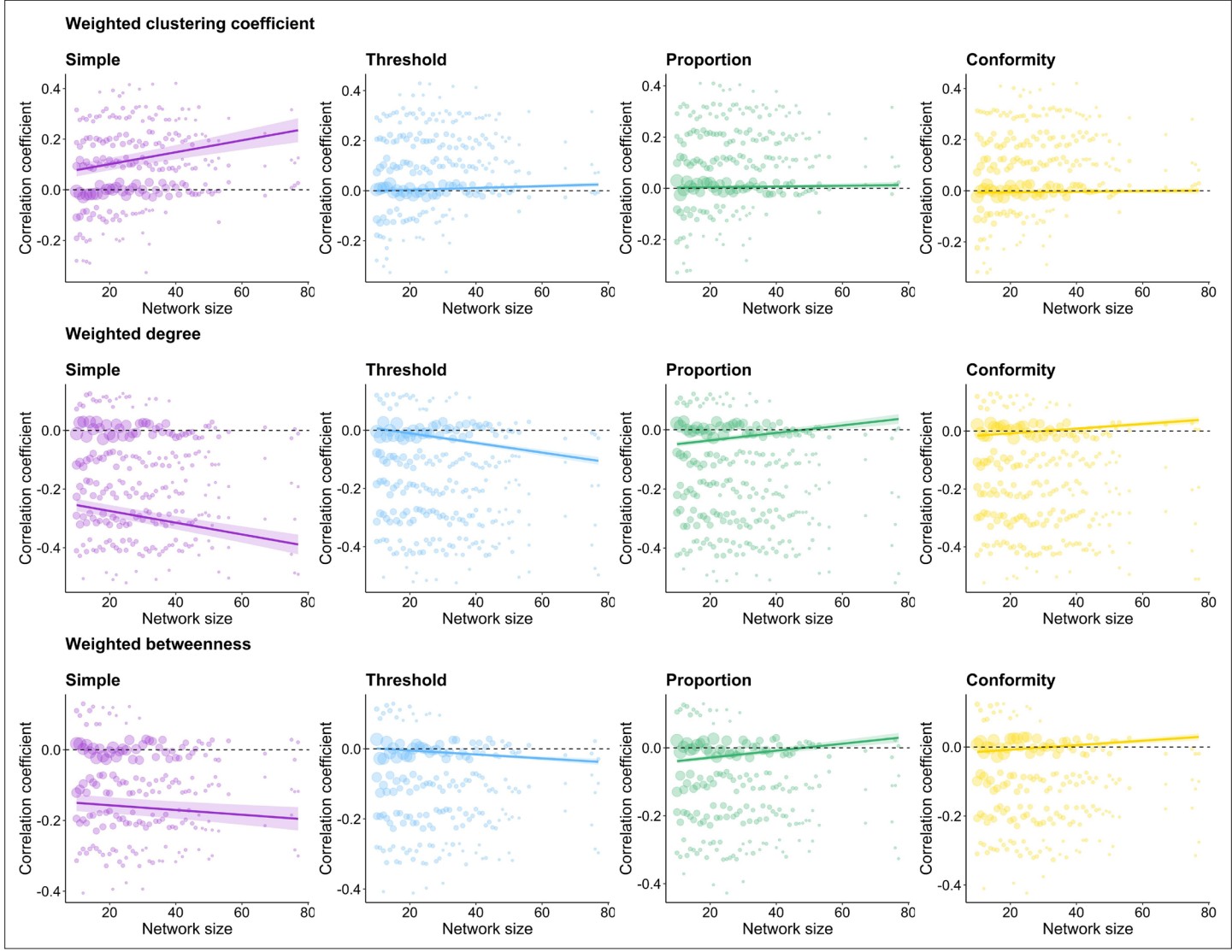

**Figure 4.** Relationship between correlation coefficient and network size across the four social learning rules. Each row shows one of the individual network metrics (top to bottom: weighted clustering coefficient, weighted degree, and weighted betweenness) and each column a different social learning rule (left to right: simple, threshold, proportion, and conformity). Average correlation coefficients across the 100 simulations per network are plotted as count dots (larger dots indicate more values for the respective value), lines represent the predicted effects generated from linear mixed-effect models (LMM) and ribbons represent the 95% confidence intervals (see *Supplementary file 1b* for model results).

The online version of this article includes the following figure supplement(s) for figure 4:

**Figure supplement 1.** Relationship between correlation coefficient and network size across the four social learning rules under different social learning rates.

**Figure supplement 2.** Relationship between correlation coefficient and network size for the threshold and conformity learning rule under different parameters.

under a strong transmission rate (i.e. *s* = 10; *Figure 4—figure supplement 1*). For the threshold model, the direction of the relationship did not change across different social transmission rates, that is correlation coefficients became on average more positive (weighted clustering coefficient) and negative (weighted degree and weighted betweenness) with increasing network size (*Figure 4—figure supplement 1*). However, the relationship was strongest (i.e. steepest slope) for larger transmission rates (*Figure 4—figure supplement 1*). Similar patterns are present for weighted degree and weighted betweenness under the proportion and conformity model where correlation coefficients increase with increasing network size (*Figure 4—figure supplement 1*). For the weighted clustering coefficient, the slope indicated opposing directions for different transmission rates under

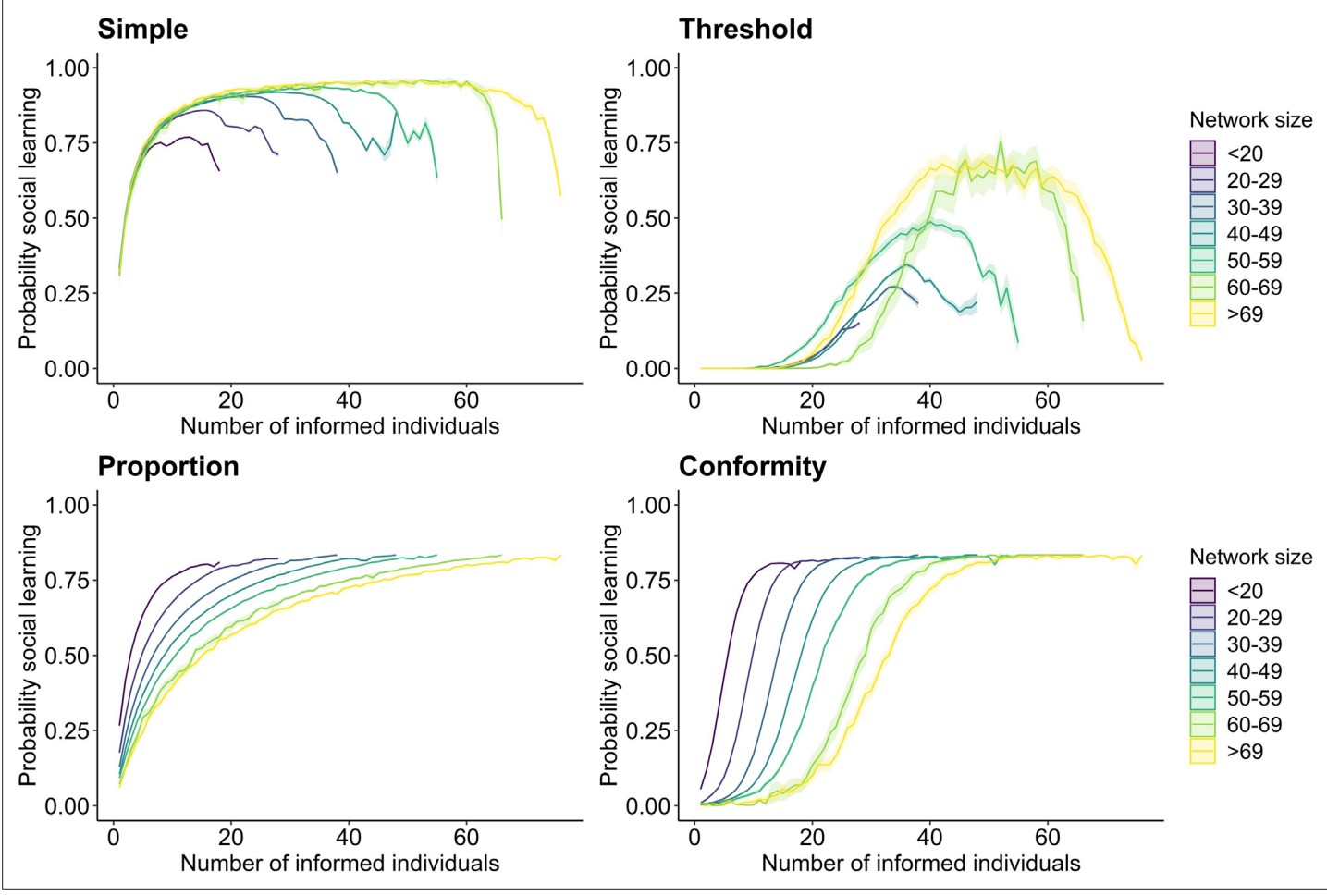

**Figure 5.** Relationship between the probability of an individual socially adopting the seeded behaviour and the number of informed individuals within the network. Panels show the results for each of the four social learning rules (simple, threshold, proportion, and conformity). The *x*-axis describes the number of informed individuals within a social network. The simulations are set so that at each timestep a new individual adopts the behaviour, whereby each time each individual has a probability of adopting the behaviour through social learning (*y*-axis) given the set learning rule. Lines plot the average probability for each timestep and ribbons show the 95% confidence intervals from the 100 simulations across the binned groups for different network sizes. Colour represents network size with darker colour indicating smaller networks. The social transmission rate, the threshold location, and the frequency dependence parameter were set to 5.

The online version of this article includes the following figure supplement(s) for figure 5:

**Figure supplement 1.** Relationship between the probability of an individual socially adopting the seeded behaviour and the number of informed individuals within the network under different social transmission rates.

**Figure supplement 2.** Relationship between the probability of an individual socially adopting the seeded behaviour and the number of informed individuals within the network under different threshold location (*a*) and frequency dependence (*f*) parameters.

the proportion and conformity model but were generally small and non-significant (***Figure 4—figure supplement 1***). The relationship between correlation coefficient and network size under the threshold model was also modulated by the threshold location '*a*' (***Figure 4—figure supplement 2***). Here, the relationship between correlation coefficient and network size was strongest for small threshold locations (***Figure 4—figure supplement 2***). The frequency dependence parameter had no effect on the predicted relationship (***Figure 4—figure supplement 2***).

## Probability of social spread

Across the different social learning mechanisms, the average probability that an individual socially adopted the seeded behaviour increased with increasing timestep (i.e. order of acquisition, ***Figure 5***). This is because with an increasing number of individuals becoming knowledgeable, the probability

to be connected to informed others and thus to reach the set learning criterion increased (*Figure 5*). Behaviours were most likely to socially spread under the simple and proportion model (on average a non-zero probability to socially spread, *Figure 5*). That is because both rules did not require to surpass a given threshold (such as set for the threshold rule or reaching the majority of an individual's connections as set for the conformity rule). Therefore, one connection to an informed conspecific was sufficient for the seeded behaviour to spread socially. However, under more complex spreading processes, social learning was limited (*Figure 5*). This was particularly the case for smaller networks when transmission followed the threshold rule. For instance, in networks consisting of less than 20 individuals, the set threshold for socially adopting the behaviour was almost never met (average probability of social learning close to 0; *Figure 5*, Threshold). The high rate of asocial learning presumably led to no clear relationship between individual network metric and order of acquisition (*Figure 2*, Threshold; see also section on '*Relationship between individual social behaviour and order of acquisition*'). For the conformity rule, larger networks limited social spread, but only at the initial phase of transmission (*Figure 5*, Conformity). In larger social networks, individuals have on average more social connections and thus at the initial stages of behavioural spread, it is less likely that the majority of an individual's social connections are already informed.

For both the simple and threshold model, the mean probability of social spread initially increased but then dropped towards the later adoptions (*Figure 5*). This may be caused by a few individuals that are not well connected in the social network, and thus have in general a low probability to socially learn under the simple and threshold rule. For instance, some great tits may only be connected to one other individual and thus, even if the number of knowledgeable individuals in the social network increases over time, it is very unlikely to reach the required threshold for social learning. For the proportion and conformity rule, the average probability of social learning peaked after a given percentage of individuals being knowledgeable (*Figure 5*).

The probability to socially adopt a behaviour was modulated by the model parameters chosen. Across all four models, the probability of social learning increased with an increasing social transmission rate '$s$' (*Figure 5—figure supplement 1*). Further, for the threshold model, the likelihood of social spread decreased with increasing threshold location '$a$', especially in smaller networks (*Figure 5—figure supplement 2*). For the conformity model, changing the frequency dependence parameter '$f$' did only slightly affect the probability of social learning in larger networks whereby with higher parameters social spread at the initial stage was more limited (*Figure 5—figure supplement 2*).

## Discussion

Using simulations on large numbers of empirical great tit social networks, we show how the underlying social learning rules individuals employ strongly influence the transmission pathways of behaviours across social networks derived from real-world data. Under learning rules that rely purely on the extent of social connections to informed others, we found that individual great tits with a higher weighted degree and betweenness, and lower clustering coefficients were likely to adopt the seeded behaviour faster, in line with common expectations of the benefits of sociality for gaining information. However, if the likelihood of adopting a behaviour depended on the ratio of connections to informed and uninformed others, such as conformist learning, social connectivity was not strongly related to the order in which individuals acquired the seeded behaviour. Notably, this contrasts with the widely proposed prediction that more social individuals may be more likely to adopt new information. Thus, our results show how the relationship between individual sociality and behavioural acquisition can change fundamentally with the type of social learning mechanism at play. Finally, we reveal that the probability of social spread under certain social learning rules is predicted to be limited in certain real-world settings, particularly in networks of a very small or large size.

Individuals differ in the quantity and quality of social connections to others which impacts several aspects of life history (*Alberts, 2019*; *Beck et al., 2021*; *Farine and Sheldon, 2015*; *Formica et al., 2012*; *MacIntosh et al., 2012*; *McDonald, 2007*), including the social transmission of information (*Aplin et al., 2012*; *Kulahci and Quinn, 2019*). Generally, individuals that hold many social connections to others, and occupy central network positions, are expected to be more likely to acquire information (*Aplin et al., 2012*; *Claidière et al., 2013*; *Hoppitt and Laland, 2011*; *Kulahci et al., 2016*). In line with these assumptions, we find that great tits with lower weighted clustering coefficients, and higher weighted degree and betweenness were more likely to adopt the seeded behaviour faster

when the underlying spreading mechanism depended on the extent of connections to knowledgeable others (*Figure 2*). However, when rules were more complex, for example when behavioural adoption depended on the ratio of connections to knowledgeable and naive individuals, sociality was not strongly related to the order of acquisition (*Figure 2*), matching the expectations from studies on complex contagions in humans (*Guilbeault et al., 2018*). For instance, in humans individuals with more social connections require stronger exposure to identify 'useful' information from the noise received from all their associates (*Hodas and Lerman, 2014*; *Hodas and Lerman, 2012*) and individuals with fewer social connections may utilize information sooner (*González-Avella et al., 2011*). These findings suggest that, under certain spreading mechanisms, the extent of social connections can reduce social spread. Thus, the concept that more social and central individuals are especially important in acquiring and subsequently spreading information (*Kulahci and Quinn, 2019*) cannot be generalized across different types of behaviours, and may even lead to erroneous conclusions. For instance, if no evidence is found that individual sociality is related to the probability of behavioural adoption, one might be led to conclude the absence of social learning where, in fact more complex social learning rules may be in operation.

These findings have important consequences for our understanding of the relationship between sociality and behavioural acquisition, and also for what may constitute an 'optimal' social structure for efficient social transmission (*Cantor et al., 2021*; *Pasquaretta et al., 2014*; *Romano et al., 2018*). In many social structures, individuals differ in their social connectivity, ranging from highly to less connected individuals (i.e. heterogenous degree distribution). In networks with higher variation in connectivity, simple contagions may spread more efficiently because highly connected individuals can act as 'hubs' (*Evans et al., 2020*; *Xue et al., 2020*). In contrast, more complex contagions may spread more slowly on networks with heterogenous degree distribution (*Evans et al., 2020*; *Xue et al., 2020*). Many animals, live in fission–fusion societies (*Amici et al., 2008*; *Silk et al., 2014*; *Wilson et al., 2014*) where individuals frequently join, leave, and rejoin groups which can result in large individual differences in social connectivity (*Sah et al., 2018*). In contrast, some animals form highly stable groups (e.g. many primates and carnivores; *Kappeler and van Schaik, 2002*; *Holekamp et al., 2007*) with lower individual variation in social connectivity (*Sah et al., 2018*). In social networks with heterogenous degree distribution, an individual's social network position may, under certain behavioural contagions, have a strong impact on the probability of behavioural acquisition whereas in groups with homogenous degree distribution, an individuals' position within the network may not be as important. Ultimately, the 'optimal' social structure for information transmission will highly depend on the behaviour and the underlying learning mechanism. For future work it will be interesting to test how different behavioural contagions spread on social networks of different species, ideally with contrasting social structures.

Our results have implications for our understanding of the costs and benefits of individual sociality. While increased access to information is one of the postulated key benefits of sociality, simply holding more connections to others may in fact hinder the adoption of novel behaviours under certain social learning rules. For instance, if a novel behaviour follows a conformist learning mechanism, highly social individuals with lots of connections may be exposed to the new behaviour sooner than less social individuals because they are more likely to be connected to at least one informed conspecific, but despite this will be less likely to adopt the novel behaviour if it requires the majority of their social connections to become informed first (*Firth, 2020*) or if many social connections make the detection of useful information more difficult (*Hodas and Lerman, 2014*; *Hodas and Lerman, 2012*). In addition, our results suggest that occupying more peripheral network positions may be costlier in terms of behavioural adoption than occupying central network positions is beneficial, but that this is dependent on the learning rule. For instance, the proportional rule and conformity rule did not show that individuals with lower social connectivity were likely to adopt very late, yet the simple and threshold learning rule showed that the average network centrality of late adopters was very low (after approximately 75% of individuals being informed; *Figure 2—figure supplement 1*) but remained relatively unchanged for earlier adoptions (before 75%, *Figure 2—figure supplement 1*). This suggests that in various cases, poorly connected individuals adopt the behaviour last and have in general a very low probability of socially adopting behaviour (see decrease in social learning probability for simple and threshold rule, *Figure 5*) but that this can be negated when certain social learning rules (e.g. the proportional and conformity rules) are in play. We speculate that the relationship between network

position and order of acquisition at the initial stage (i.e. before approx. 75%), and the large variation in correlation coefficients (*Figure 3*) is also highly determined by the demonstrators' network position (i.e. the starting positing of spread, *Banerjee et al., 2013*) and the underlying network structure (*Cantor et al., 2021*; *Evans et al., 2020*; *Romano et al., 2018*) which may warrant future work. Finally, how biologically meaningful differences in the order of acquisition are will ultimately depend on the behaviour and context, and the actual observed variation in adoption times. Therefore, quantifying real individual variation in the timing of behavioural acquisition in the wild will be crucial for our understanding of the potential costs and benefits of individual sociality.

Examining behavioural innovation and its subsequent spread in nature is challenging (*Klump et al., 2021*; *Whiten and Mesoudi, 2008*). This is because new behaviours are often only detected once the majority of individuals in a population are already knowledgeable. Alternatively, behavioural innovations may occur much more frequently but remain undetected if the behaviour does not spread far (e.g. because the behaviour is mechanically challenging for individuals [*Gajdon et al., 2006*] or when the carryover of older, outdated behaviour hinders the spread of novel and more adaptive behaviours [*Aplin et al., 2017*; *Barrett et al., 2019*]). Here we used simulations on real-world empirical networks, and demonstrate that the ability of a behaviour to socially spread depends on its underlying social learning mechanism. Our findings are thus consistent with other studies that investigated different spreading processes (*Cantor et al., 2021*; *Evans et al., 2021*; *Evans et al., 2020*; *Nunn et al., 2009*). For instance, *Evans et al., 2020* simulated a simple and conformity contagion on different social structures and show that disease/information spreads faster (i.e. time until a certain number of individuals had been infected/informed) under a simple contagion compared to a conformity contagion. Our study supplements past research by specifically focusing on the diffusion dynamics on the individual rather than the population level. Across the four models, the probability that individual great tits socially adopted the seeded behaviour increased with increasing timestep (*Figure 5*). This is expected as at each timestep in simulations a new individual becomes knowledgeable and remains in this state thereby increasing the number of knowledgeable individuals within the network. However, social transmission under certain social learning mechanisms was limited. For instance, behaviours that needed to surpass a given threshold of social connections to informed others (i.e. threshold or conformity rule) required more asocial learning events in the initial spreading phase to be able to subsequently transmit via social learning (*Figure 5*, *Figure 4—figure supplement 2*). In contrast, for the simple and proportion rules, the probability of social transmission was always higher, even in the initial spreading phase and when considering different transmission rates (*Figure 5*, *Figure 5—figure supplement 1*). This demonstrates that in the initial stage some behaviours may have a higher likelihood to spread socially, whereas other behaviours, following more complex processes, may rely on a larger extent of asocial learning events or social reinforcement. Therefore, the majority of empirical research on behavioural spread in animals may in fact examine more simple social transmission processes (because they are easier to detect and observe) which may bias our picture of the existing social learning mechanisms of animals and behavioural innovations.

We found that network size impacted behavioural spread across the four transmission models. The strength of the relationship between individual network metric and the order of acquisition increased with increasing network size under the simple and threshold rule (*Figure 4*) and was mediated by the social transmission rate (*Figure 4—figure supplement 1*) and the threshold location (*Figure 4—figure supplement 2*). The weaker correlation between individual network metrics and order of acquisition in smaller networks might be caused by the relatively reduced likelihood of social spread (*Figure 5*). For instance, for the threshold rule, individuals in larger networks have an overall higher number of social connections (compared to individuals in smaller networks) which facilitates reaching the set threshold for social learning and thus increases the likelihood of social spread. This is supported by our results when simulating spread using different threshold values where behaviours were most likely to socially spread across different network sizes under small thresholds (see *Figure 2—figure supplement 5*). In addition, our findings may suggest that in smaller networks, the order of acquisition is mainly predicted by who an individual is connected to (e.g. whether it is directly connected to the demonstrator) compared to the number and extent of social connections it has. In contrast, in larger networks, the probability of behavioural acquisition may be strongly influenced by the number and extent of connections an individual has. Therefore, the importance of 'who' you know versus 'how many you know' may differ in networks of varying size. Past studies have investigated behavioural

spread on relatively large and static networks (using simulations: *Cantor et al., 2021*; *Evans et al., 2021*; *Nunn et al., 2009*; *Voelkl and Noë, 2008* or natural observations: *Allen et al., 2013*; *Aplin et al., 2015a*). However, in many species social associations can be highly dynamic and only the social connections at a relatively small temporal scale (e.g. at the time of emergence) may predict an individuals' decision to adopt a novel behaviour. Our findings that network size can impact behavioural spread thus have important consequences for our understanding of the influence of wider society structure on when and where behaviours may emerge, and how to interpret empirical results. As such, it is important to improve our understanding of the factors that give rise to different social network sizes and structures and to consider networks on an appropriate spatiotemporal scale or dynamic versus static networks (*Hasenjager et al., 2021*; *Hobaiter et al., 2014*). For instance, spatiotemporal variation in environmental features (such as the availability and distribution of resources) may influence population densities and subsequently local social network size and structure across space and time, and as such may influence local social spread.

Our study suggests further questions for future research. Our models assume that each individual adopts a seeded behaviour under the same 'learning rule'. While variation in individual learning rules and their impact on behavioural contagions have been widely examined in humans (*Aral and Nicolaides, 2017*; *McCullen et al., 2013*; *Melnik et al., 2013*; *Muthukrishna et al., 2016*), its' investigation in animals remains scarce. However, individuals may differ in the extent of social information use and the thresholds required for behavioural adoption (*Chimento et al., 2022*), which may be context and state dependent (*Penndorf and Aplin, 2020*; *Rendell et al., 2011*). For instance, dominance rank (*Krueger et al., 2014*) and sex (*Aplin et al., 2013b*) have been shown to be related to variation in social information use and individuals may only copy behaviour from certain individuals, based on familiarity or kin (*Boogert et al., 2018*; *Kavaliers et al., 2005*). Therefore, learning rules may differ within individuals (e.g. with changes in age or dominance) and between individuals (e.g. sex, relatedness). Future simulation and empirical studies could explore how heterogeneity in learning rules, within and between individuals, and variation in acquisition versus adoption, impact information flow (*Chimento et al., 2022*). In addition, we only explored learning rules which might be relevant for our study system. However, future research could test the same and different rules on species with different social structures. For instance, in species with more stable social groups such as in primates and many carnivores (*Holekamp et al., 2007*; *Kappeler and van Schaik, 2002*), kin- or dominance-based learning rules may be more applicable.

Furthermore, using simulations did not allow us to test changes in the social network resulting from behavioural adoption: individuals may occupy more central social network positions once performing a new behaviour (*Kulahci et al., 2018*; *Kulahci and Quinn, 2019*). This can be the case if individuals preferentially associate with knowledgeable others (*Kulahci et al., 2018*), or if individuals change their behaviour in response to information acquisition which can also lead to an increase in social connections (*Kulahci and Quinn, 2019*). Such dynamics are not reflected in our study, and would require the investigation of natural behavioural spread, ideally under experimental conditions in the wild. In addition, our models assume that once individuals become 'informed', they cannot return to an 'uninformed' state. In natural conditions, however, individuals may return again to an 'uninformed' state if the novel behaviour was not rewarding. Future studies incorporating these aspects could provide further new insights into the patterns of social transmission and its link to sociality. Finally, we used empirical networks to capture the fine-scale social association patterns between wild birds and to explore how different behavioural contagions spread on them. While it is important to test predictions on real networks, such an approach also has additional considerations. For instance, network metrics are often correlated and dependent on one another (*Supplementary file 1a*) which makes it difficult to tease apart the direct effects on the probability of behavioural adoption for each metric alone. In studies using simulated networks, interdependencies can be controlled for in certain ways and the effects of each metric could be teased apart (e.g. by using sensitivity analysis). Therefore, we suggest that, while both empirical and simulation studies can provide valuable information on their own, considering both complementary to each other will improve our understanding of how behaviours spread on different networks and how different social network metrics relate independently to behavioural adoption.

From copying the majority (*Aplin et al., 2015a*; *Danchin et al., 2018*; *van de Waal et al., 2013*) to learning from specific tutors (*Canteloup et al., 2020*; *Wild et al., 2019*), individuals use a large range

of different learning strategies (*Hoppitt and Laland, 2013*; *Kendal et al., 2018*). While, an increasing number of studies show how social network structure can influence social transmission (using simulations: *Evans et al., 2021*; *Romano et al., 2018*; *Voelkl and Noë, 2008* and empirical data: *Firth et al., 2016*; *Naug, 2008*; *Romano et al., 2018*), our research highlights the importance of the underlying social learning mechanism in shaping the transmission pathways across social networks. We demonstrate that the common assumption that sociality is linked to a higher likelihood in acquiring information and adopting new behaviours cannot be generalized for behavioural spread in real-world networks. This also sheds new light on our current understanding of the costs and benefits of individuals sociality and asks to focus more on the social learning mechanism at play, and to differentiate between access to information and behavioural adoption (*Chimento et al., 2022*). In addition, we reveal that social transmission can be limited under certain adoption rules (such as the threshold rule), and social networks of particular size. Our findings thus have important consequences for our understanding of whether and how behaviours spread across different social networks, and subsequently the establishment of traditions and cultures. Further, differences in spreading mechanisms alter predictions of what may constitute optimal social structures for the transmission of information, and how selection may act on sociality.

## Acknowledgements

We thank the large number of contributors to the data collection, and we are very grateful to Will Hoppitt for all his input. The study has been supported by grants from NERC (NE/S010335/1 & NE/V013483/1), ERC (AdG 250164), and BBSRC (BB/S009752/1). We also thank three anonymous reviewers and the Editor for their comments and suggestions on this manuscript.

## Additional information

### Funding

| Funder | Grant reference number | Author |
|---|---|---|
| Natural Environment Research Council | NE/S010335/1 | Ben C Sheldon |
| European Research Council | AdG 250164 | Ben C Sheldon |
| Biotechnology and Biological Sciences Research Council | BB/S009752/1 | Josh A Firth |
| Natural Environment Research Council | NE/V013483/1 | Josh A Firth |

The funders had no role in study design, data collection, and interpretation, or the decision to submit the work for publication.

### Author contributions

Kristina B Beck, Conceptualization, Formal analysis, Visualization, Methodology, Writing – original draft; Ben C Sheldon, Conceptualization, Supervision, Funding acquisition, Project administration, Writing – review and editing; Josh A Firth, Conceptualization, Supervision, Visualization, Methodology, Writing – review and editing

### Author ORCIDs

Kristina B Beck ⓘ http://orcid.org/0000-0002-5027-0207
Ben C Sheldon ⓘ http://orcid.org/0000-0002-5240-7828
Josh A Firth ⓘ http://orcid.org/0000-0001-7183-4115

### Ethics

All work was subject to review by the University of Oxford, Department of Zoology, Animal Welfare and Ethical Review Board (approval number: APA/1/5/ZOO/NASPA/Sheldon/TitBreedingEcology).

Data collection adhered to local guidelines for the use of animals in research and all birds were caught, tagged, and ringed by appropriate BTO licence holders.

## Decision letter and Author response
Decision letter https://doi.org/10.7554/eLife.85703.sa1
Author response https://doi.org/10.7554/eLife.85703.sa2

## Additional files

### Supplementary files
• Supplementary file 1. Supplementary tables. (a) Correlation coefficients between the three individual network metrics. Shown are the correlation coefficient between the network metrics weighted clustering coefficient, degree, and betweenness. (b) Effects of network size on the mean correlation coefficient. Shown are the effects of network size on the mean correlation coefficient for weighted clustering coefficient, degree, and betweenness for each of the four social learning models. We report the estimate, the standard error (SE), test statistic ($t$), and p values.

• MDAR checklist

### Data availability
All data and code to reproduce the analyses can be accessed at https://osf.io/6jrhz/.

The following dataset was generated:

| Author(s) | Year | Dataset title | Dataset URL | Database and Identifier |
|---|---|---|---|---|
| Beck K | 2022 | Data and R code for: 'Social learning mechanisms shape transmission pathways through replicate local social networks of wild birds' | https://osf.io/6jrhz/ | Open Science Framework, 6jrhz |

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
