## [Editor Report]

This valuable study will be of interest to researchers in the fields of behavioural ecology, social ecology and evolution, and network science. The authors use simulations on empirically-recorded great tit social networks to examine how behavioural contagion might spread through social groups if individuals follow different social learning rules. The evidence supporting the conclusions is convincing, with careful modeling and parameterization for the chosen system.

---

## [Decision Letter]

**Decision letter after peer review:**

[Editors’ note: the authors submitted for reconsideration following the decision after peer review. What follows is the decision letter after the first round of review.]

Thank you for submitting your article "Social learning mechanisms shape transmission pathways through replicate great tit social networks" for consideration by *eLife*.

Your article has been reviewed by three peer reviewers, and the evaluation has been overseen by a Reviewing Editor and a Senior Editor. The reviewers have opted to remain anonymous.

Comments to the Authors:

All three reviewers found the modeling approach and main results valuable. That said, they raised a number of major concerns, which can be summarized as follows (for many additional points, please see their original reports below):

(1) A more thorough description of the model should be provided. Ideally, all code should be made available so that readers can replicate the modeling.

(2) Parameter selection is weakly justified, and sensitivity analyses for the parameters are missing.

(3) It is unclear to what extent the results depend on the characteristics of great tit networks and how they are relevant to other species. Related to this point, the relevant literature needs to be covered better, including earlier work using empirically-recorded networks to simulate diffusion processes.

*Reviewer #1 (Recommendations for the authors):*

In this manuscript, the authors simulate learning processes using data collected from wild birds on their social networks. The results suggest that the nature of the learning process, as well as the size of the social network, determine which individuals will adopt a new behavior faster.

1. The main advance is simulating information spread over empirical networks rather than artificial networks. I doubt that this constitutes a major advance. Specifically, the authors mention in the discussion that the results are in agreement with previous results obtained from artificial networks. Therefore, while it is important and interesting to use empirical networks, their usage did not seem to provide additional insight.

2. The results depend on the characteristics of great tit networks. Although many different networks were used in the simulations, we can expect different results if networks of other species would be used. It would be helpful if the authors can provide some information about the structure of the empirical networks, and perhaps some predictions regarding their relevance to other species.

3. As far as I could tell, the specific code and data for running these simulations were not shared (not just the package). This makes it impossible to replicate the results and assess the data structure.

4. Learning rules in the wild were briefly mentioned in the introduction. Is there any evidence for the rules being used in wild animal populations, supporting the inclusion of these four rules?

5. Figure 2 and S2: I don't really understand why the order of acquisition is plotted as predicting the network traits (e.g., clustering coefficient). If there is any connection between the two it should be the other way around. To me it makes these figures confusing. Perhaps related, at least in the lower panels all individuals seem to have very similar trait values (e.g., weighted degree) -- how can this be explained?

6. L. 94-100: Earlier, the authors mention that individuals may tend to learn from specific individuals such as relatives or with a given trait value. To me that sounds like a very likely mechanism, such as "learn only from your mother, or from your strongest affiliation". I would be very interested to see versions of such rules tested here.

L. 90: Did you mean predator avoidance?

L. 152: The introduction mentioned weekly networks but here weekends are reported. How much time does each network include?

Figure 1: This figure is referenced only from the methods section. It raises a question: Why were the networks generated from only one location each time? I assume that two birds may have also interacted in adjacent locations. Perhaps some data on the overlap across adjacent locations can help.

L. 206-207: This statement does not seem to be supported by Figure 3, in which the correlation coefficients for weighted degree and weighted betweenness are actually lower in the simple learning rule.

L. 248…: I was surprised to see here, in the last part of the results, an explanation of the simulation procedure, which belongs either to the methods section, or to the beginning of the results.

L. 470: foal should be focal

*Reviewer #2 (Recommendations for the authors):*

Overall, I found the article interesting and feel that it provides an interesting examination of the consequences of how individuals learn on how behavioural contagions spread through animal societies. Previous studies have not applied these types of model to empirically-derived animal network data or studied as many forms of behavioural contagion simultaneously. Consequently, while (in the context of broader network science research) the individual results here are not new, it is valuable to see them presented together and in a way that is accessible to behavioural ecologists.

I felt the article was well-framed in the introduction, and the results were largely presented clearly and intuitively. In particular, I thought that Figures 2 and 5 were an excellent way to clearly show some of the main results of the study. However, there was insufficient information provided about the models or the experimental design of the modelling component prior to the results, which made interpretation of model results harder than it should be and at times made the results hard to follow when intertwined with methodological information.

I have some concerns about the modelling approach used, although perhaps some of them are related to the rather limited information provided on the simulation approach (I would really encourage the authors to provide clearer methods that incorporate functions for the learning rules and a step-by-step description of the simulation algorithm). It is not clear to me whether acquisition is deterministic based on likelihoods or probabilistic, and the step explaining how the likelihood of information being acquired socially or asocially is particularly unclear. This plays in to some concerns about the appropriateness of a strictly order-based algorithm -- it seems a rather artificial choice based on an existing statistical model rather than clearly biologically justified. Given the shape of the adoption curves for different contagions will differ, and that there is a probability of asocial component, avoiding a time-based approach seems like it could potentially have different consequences for different social learning rules. I can potentially see an argument that individuals would never truly acquire information at the "same" time, but this perhaps raises the question as to: (a) how meaningful changes in behaviour might or might not be to the subsequent individual to acquire the behaviour; and (b) how biologically meaningful differences in adoption time are (outside of very particular biological contexts such as during a predation event).

Another concern of the work presented here is that there are minimal checks of sensitivity to model parameterization (aside from changing the threshold number of connections in the threshold model). Parameter selection for the different forms of social learning is fairly weakly justified and not well explained so it is not clear what effect this might have on the generalizability of the results. For example, previous modeling work has suggested that the difference (in speed) in how simple contagions and contagions based on conformist learning spread through some types of networks depends on how easy the contagion can spread. This may not impact results related to the order of acquisition, but it is hard to tell if this is likely to be the case, especially given there is also an asocial learning component that has an impact on the adoption of behaviours.

I also found it interesting that the networks were weighted using simple ratio indices, and the reasoning for this wasn't clear. There are some important assumptions hidden in this choice that could potentially impact the results of the study related to what these indices relate to and how individuals learn. An (intentionally) naïve starting point (in my mind at least) would be that the likelihood of social learning depends on the number of times individuals associate at a food resource rather than a proportion of times that they occur together (e.g., individuals that occur together 4 out of 10 observations may well have more opportunities to learn than 1 out of 2 times). This is clearly a very direct interpretation of the simple ratio indices which ignores the potential for associations elsewhere and their "representative value". It is also based on the assumption that number of associations is the factor that drives social learning (as opposed to, e.g., associations, within a set amount of time, strength of social bond, etc.). However, it would be good to more clearly set out the reasoning for this choice and perhaps test or discuss the sensitivity to different assumptions here.

Away from the research itself, one frustration when reading this paper is that it does a fairly poor job of placing itself in the context of the wider literature. First, while it does a good job of citing the relevant studies that conduct similar modelling work in animal societies, there is relatively little effort to engage with the findings of these studies in the introduction or discussion of the paper. There is a real opportunity here to unpack the results of this study in relation to similar and contrasting findings from other papers that is missed here. Different papers have focused on different aspects of how social structure and connections influence contagions in animal societies and by linking better with some of these papers it could perhaps address how the findings of this study might generalize (e.g., to different social structures, considering different "transmissibilities" of contagions, etc.). Second, there is little effort made to acknowledge or consider the large number of modelling studies that address similar questions in the broader network science literature. While the network here is empirically derived, from that point on this study is purely computational and there are studies that have addressed very similar or overlapping questions elsewhere in this literature (e.g., how the number of connections influences speed of acquisition for different forms of contagion).

Related to this point, it would be nice to see a more nuanced discussion about the strengths and weaknesses of computational research that either applies simulation models to a single empirical case study vs. that the applies similar computational models to more generalized network structures. While there was a point when these types of model were applied to very generic network structures (random, small-world, etc.) and to an extent still are in network science (where the research aims are somewhat different), more frequently now studies that use simulated network structures do so with express biological questions in mind and design simulated networks accordingly. Taking this approach is a powerful way of tackling specific questions and/or generating a range of generalizable structures. Equally applying these types of models to particular empirical case studies is very valuable in its own right for different reasons. Related to the previous point, I think it would be great to make the most of their complementary strengths to better integrate the lessons learned from these different approaches.

This recently published paper is perhaps relevant/useful:

https://royalsocietypublishing.org/doi/full/10.1098/rspb.2022.1001

L20: Given the weak correlations illustrated in Figure 3, it feels slightly misleading to describe this as a "strong" relationship.

L22-23 (and elsewhere): Discussion of this idea throughout the paper doesn't acknowledge previous work showing this outside ecology. This review contains more links to studies in network science as a useful resource https://onlinelibrary.wiley.com/doi/full/10.1111/oik.07148?saml_referrer. For example, this paper https://journals.plos.org/plosone/article?id=10.1371/journal.pone.0020207 tackles how conformist social learning leads to this pattern.

L48: "requiring exposure to multiple sources" isn't necessarily a difference between infection and behaviour spread in networks (e.g., see literature on dose-response curves).

L76-78: Evans et al. (2021) also consider a simple contagion and conformist learning and explore the potential implications of considering the simple contagion as being something other than the spread of infection in the discussion.

L83-85: While I think a really interesting aspect of this study is its exploration of the role of network size, I am not sure how well this criticism works (in its current form) given that even big networks can capture interaction patterns at small spatial scales, and that how network structure depends on the temporal scale will be highly study-system dependent.

L96-100: Given the methods come last, these rules need to be more accurately described here to help with the interpretation of the results. It would also be good to provide a more in-depth exploration of previous theory/discussion about these rules beyond ecology.

L112-113: Low clustering coefficient will not always indicate a less sociable individual, this will depend considerably on the broader structure of the network. It would seem to in this study, but some greater context would be helpful here. In the results it would be helpful to quickly describe the correlation between the centrality measures used in the studied networks.

Figure 1: Would be useful (and take up no extra space) to give the thresholds used here.

L151-154: It would be good to provide more clear information on networks per feeder site, network appearances per bird, etc., and an indication of what timespan were data included from. Also perhaps valuable to point out that the minimum of 10 (and rest of these descriptive stats) applies after some networks were excluded.

Figure 2: One thing I found particularly interesting in these results is that for simple, threshold and to some extent conformity there appears to be a stronger pattern of being the most peripheral is costly rather than being the most central is beneficial. Clearly, this may depend on what the information/behaviour that is spreading is related to, but it's a neat result and perhaps worthy of some discussion for what it means for social ecology/evolution in this system.

L210: Is this now meant to refer to Figure 3?

Figure 3/Figure 4: Even the correlations different from zero are (predominantly) small. Later in the paper it might be interesting to discuss how biologically meaningful they are in networks of different sizes in this system. One thing apparent in these results is that there is a lot of noise (presumably) related to the seeded individual. It may be worth using this to highlight that in small networks who you know is just as/more important than how many you know -- even if it is just as a suggestion for further research.

Figure 4: I found distinguishing between the two blues here pretty difficult. Another colour scheme may be clearer or a different way of presenting the data given how dense the point clouds are.

L219-231: It would be good to make this aspect of the experimental design clearer earlier. I also found these results were written less clearly than other parts -- some rewriting might be helpful.

L248-255: Some indication of how the model broadly works such as this should ideally come at the end of the methods to help with interpreting the results in this methods-last format.

L270-274: Would perhaps be helpful at the end of the introduction to set up the model or in the methods?

L362-364: Is there not a little more nuance here given that there won't necessarily be a single learning rule for each behaviour so it suggests perhaps that the importance of different learning rules varies as a behaviour spreads.

L378-380: Similarly to the previous point, an appropriately parameterized (dynamic) network for a large population can capture interactions at very fine spatial and temporal scale -- this doesn't seem so much a point directly related to network size. One aspect of network size that perhaps becomes interesting is that the importance of "who" you know versus how many you know changes in importance in different size networks.

L383: Is there a reason for using network "shape" here rather than more established terms like topology or structure?

L387-405: Good to see a sensible consideration of model limitations.

L449-453: Is there empirical data to support this for this study system given it has been the subject of previous experimental work? Justifying with previous empirical work would strengthen this point considerably.

L449-453: What biases do you introduce to the networks by only including interactions at a single feeder? How many individuals use multiple feeders?

L470: I am not convinced the use of clique (in the strict network analysis definition) is correct here.

L505: I would suggest making it clear earlier in this description that simulations were repeated multiple times in each network.

L520: Is this number correct given different thresholds were used too for further simulation runs?

L530-53: Would be good to provide confirmatory information on model goodness of fit checks and clarify how statistical inference was done (presumably from the full/fitted model?).

I hope my comments help improve the manuscript.

*Reviewer #3 (Recommendations for the authors):*

This study is a timely theoretical exploration of how variation in transmission rules interacts with variation in social phenotype to influence diffusion dynamics. The authors predict that the likelihood of an individual adopting novel behaviors should depend on the learning rule, as well as the individual's sociality. The authors explore the spread of behavior under 4 different learning rules: a simple adoption rule, a threshold rule, a proportion rule and a conformity adoption rule. They quantify the sociality of individuals by calculating their weighted degree, clustering coefficient and betweenness.

The authors find that under simple and threshold rules, high degree, high betweenness, and low clustered individuals acquired the seeded behavior earlier in simulations. Under proportional and conformity rules, there was no strong relationship between social phenotype and order of acquisition. The authors find that network size predicts the magnitude and direction of the correlation between social phenotype and order of acquisition and that this relationship also depends on the learning rules.

Strengths

1. Overall the paper is well written, and the motivation for using computational simulations is well warranted to explore this question.

2. The topic is timely, and tackles an important theoretical question of how variation in learning rules might interact with social phenotype to influence cultural diffusions. This is a difficult topic to address but is critical for improving our understanding of how diffusions might differ between populations.

3. The authors construct simulations using real social networks of great tits, rather than artificially generated networks, which is a rarity, and thus of great value, in the SL modeling literature. These networks are hard earned -- taken at a relatively fine temporal scale from weekly sampling.

4. The authors provide a thorough discussion of the implications of their results.

Weaknesses

1. One main weaknesses of the paper is the lack of details given about the transmission model. The authors do not provide equations, descriptions of parameters, a detailed schedule of the model. The descriptions they do provide are spread throughout the manuscript, making it more difficult to assess. NBDA is definitely an appropriate model for the question they want to answer, but it seems like the authors have altered some features of the model (e.g., only 1 individual can learn per timestep, 1 individual must learn per timestep). This lack of clarity makes it harder to assess the results they present. Further, from their description, it appears that they have allowed for asocial learning, which adds unnecessary noise to a study that is focused on social transmission.

2. Another main weakness is that the authors do not use a sensitivity analysis, and thus it is difficult to assess the relative effects of each network metric, as they are not necessarily independent of one another. For example, degree and clustering can be correlated simply as a result of how clustering is calculated. This is the downside of using real networks, as without synthetic data, there may be insufficient data to perform a sensitivity analysis. Further, the authors do not present an assessment of variation in their results, instead showing mean values within network size as evidence of their claims.

3. Related to the interpretation of sociality, there is opportunity to increase clarity. The authors describe more social individuals as having a high degree, high betweenness, and low clustering, and less social individuals as low degree, low betweenness and high clustering. One could also imagine a bird who has high degree, low betweenness, and high clustering, being at the center of their group, but rarely going between groups. It seems harder to argue that this bird is less social than a bird with high degree and high betweenness but low clustering. The manuscript would benefit from a careful description of how different combinations of these social metrics could be interpreted.

Point by point comments

1. It should probably be mentioned somewhere in the introduction that social learning rules apply to either the social transmission of novel behavior (e.g., Aplin et al. 2015) or the social influence of others on behavior (e.g., Pike and Laland 2010, Danchin 2018), and that you aim specifically to look at social transmission.

2. The initial conditions of the simulations are not well enough explained before we get to the results. I was left wondering how the authors chose the first knowledgeable agent, which isn't answered until later.

3. The methods section could use more explanation. Those who are unfamiliar with NBDA would need to refer to other publications to see the equations, especially the meaning of 's=1', etc. Also, what are the other parameters set to (λ, A)? Consider including the equations, as well as a more thorough description of parameters. The same could be said for equations describing the network metrics.

4. Related to point 4, what happens when A=0, in a pure social learning environment? This would reduce stochasticity due to asocial innovations, and would provide a pure test of the effect that authors predict arises from sociality and learning rules.

5. L470 "foal" should be "focal".

6. I actually think it's more helpful to show results when you standardize network size in the main text, and put Figure 2 in the supplement. Figure 2 is difficult to read, and something like Figure S2 is easier to interpret if you're assessing relative differences in diffusion dynamics. Also rather than presenting each network size as a color, select one network size (or a binned size) and present a variation metric (e.g., percentile intervals).

7. Suggest changing section title "Social network size and behavioral spreading" to match first section "Relationship between…". Also suggest "diffusion" rather than behavioural spreading. After reading the section, this seems more to do with how network size impacts the correlation between variables, rather than the diffusion itself. Maybe change the heading to reflect this?

8. L204 – 231: Overall I think this section is fairly dense compared to the first section, and after reading it several times, I'm still not sure what I should take away from it. It looks like you have a very low N at large network sizes, which could drive some of these correlations in Figure 4. The fact that agents can asocially learn also makes it hard to interpret what these correlations mean.

9. L204-216: I had to read the beginning of this section several times, and it's more confusing than the first section of results. The results communicated until L210 do not relate to network size, and seem to repeat the previous section. I suggest removing this or incorporating it into the previous section and starting with how network size affected simulation dynamics.

a. Also I suggest rewriting to avoid putting the variable of interest in parentheses (e.g., L 209, 213).

b. L208: "the mean average network metric" was confusing -- do you mean "average network measure"?

c. L210: I suggest "The direction and magnitude of the correlation between ind. Sociality and order of acquisition were predicted by network size. This relationship was modulated by transmission rule…" to improve clarity.

10. Figure 4: I find it very hard to see all 4 lines, maybe choose different colors?

11. L224: Which means network metric?

12. L248: This information about initial conditions should come before the results.

13. L248: "Spreading simulation" -> diffusion simulation.

14. L253: Without the equations written out in the methods, it's difficult to assess how the learning model works. Is asocial learning turned off under obligate social learning? It's my understanding that in NBDA, the s parameter controls the relative strength of social learning per unit connection to asocial learning. In the usual formalization of NBDA, the probability of asocial learning is constant in all individuals, contra L251 which states that asocial learning only occurs in unconnected individuals. Does your model assume that individuals who are "well-connected" (also undefined in the manuscript) have the $A$ parameter set to zero? If this is the case, the authors should include a justification/definition of being well-connected.

15. L317: I'm still finding it hard to wrap my head around how a bird with low clustering is central and highly social. A nice way to explain/justify the differences between more and less social individuals would be to make a figure of an exemplar network, with several stereotypes highlighted, along with their social metrics.

16. L332: Cantor et al. (2021, Proc. R. Soc. B) should probably also be cited here, as they measure the performance of recombination and subsequent diffusion.

17. L342: Overall, this manuscript has synergy with the study "Cultural diffusion dynamics depend on behavioural production rules" (doi.org/10.1098/rspb.2022.1001), which explicitly explores the difference between acquisition and usage, and also uses NBDA as a generative model. It would be relevant to cite here.

18. Figure 5: If individuals have a low probability of social learning, do they have a high probability of asocial learning? Or not learning at all? Are there cases when both the probability of individual learning and social learning are low? Also, this is another case where normalizing the x axis between network sizes would be more informative. The authors might set asocial learning to 0 and simply directly measure the probability of acquisition by each naive agent at each time-step, since the manuscript is focused on social transmission rather than social transmission and asocial learning.

19. Related to the interpretation of the model, the authors use the word "adopt" throughout the manuscript, although one could argue that their model is not of adoption, but of knowledge transmission, since there is no mechanism to determine whether individuals would actually use the behavior once acquiring knowledge of it. In other places, the authors have used the language of knowledge transmission (e.g., Figure 1 caption). It might be best to stick with knowledge transmission throughout the paper.

---

## [Author Response]

[Editors’ note: the authors resubmitted a revised version of the paper for consideration. What follows is the authors’ response to the first round of review.]

Comments to the Authors:All three reviewers found the modeling approach and main results valuable. That said, they raised a number of major concerns, which can be summarized as follows (for many additional points, please see their original reports below):(1) A more thorough description of the model should be provided. Ideally, all code should be made available so that readers can replicate the modeling.

A major change to the accessibility of the model and associated methods description is the structural change of the manuscript, i.e. through the editors allowing us to present our methods section before the results, we are now able to relay this information more clearly. In our revised manuscript, we provide further description of the model, all relevant equations, and have added more detailed explanation of our methodological approach. Further, we have made all data and code available at https://osf.io/6jrhz/ so that the readers can replicate the modelling, and also easily modify it to fit their own system’s needs.

(2) Parameter selection is weakly justified, and sensitivity analyses for the parameters are missing.

We apologise for this shortcoming. In our revised manuscript, we added a justification of our parameter selection in the method section on ‘Simulations’ and we also perform a range of sensitivity analyses by testing multiple combination of different parameters for each of the different simulation types (lines 324-330; line numbers refer to the revised manuscript). In the main text, we present results on the parameters s=5 (social transmission rate), f=5 (frequency dependence parameter), a=5 (threshold location), but we also present results using a range of smaller and larger parameters in the extensive supplementary material (Figure S6-S15). Further, by providing all data and code, interested readers can now repeat our analysis and also perform them using different parameters too. Interestingly, we find that the parameter selection generally makes little difference to the conclusions drawn here.

(3) It is unclear to what extent the results depend on the characteristics of great tit networks and how they are relevant to other species. Related to this point, the relevant literature needs to be covered better, including earlier work using empirically-recorded networks to simulate diffusion processes.

We apologise for this shortcoming. In our revised manuscript, we provide a much more detailed description of great tit social structure in the methods (line 165-172, 240-247, Figure S1, S2) and elaborate more in relation to how our results on great tit networks may be similar/different to other species with similar/different social structures (lines 586-602). We also provide a range of points in relation to which considerations/findings are likely to be relevant across systems. In addition, we substantially rewrote our introduction and discussion, now including relevant work on similar topics in sociology and network science, and engage more with findings of other studies (e.g. lines 79-82, 89-96, 572-576, 641-645).

Reviewer #1 (Recommendations for the authors):In this manuscript, the authors simulate learning processes using data collected from wild birds on their social networks. The results suggest that the nature of the learning process, as well as the size of the social network, determine which individuals will adopt a new behavior faster.1. The main advance is simulating information spread over empirical networks rather than artificial networks. I doubt that this constitutes a major advance. Specifically, the authors mention in the discussion that the results are in agreement with previous results obtained from artificial networks. Therefore, while it is important and interesting to use empirical networks, their usage did not seem to provide additional insight.

We have now revised various aspects of the writing in relation to this comment. Indeed, we believe that the main advance of our study is not only the (1) use of empirical networks as identified by the reviewer here but also (2) that we test four different types of behavioural contagion simultaneously, (3) to specifically examine the relationship between individual sociality and the probability of behavioural adoption (rather than just comparing the effect of different contagions on the ‘efficiency’, e.g. the speed, of spread within a population), and (4) to explore the effect of network size by considering real networks generated from social behaviour on a relatively small temporal scale. While some of these aspects may have been investigated in human and network sciences separately, they have not been investigated in tandem nor have they been explored in animals, and the availability of such fine-scale information on the social behaviour of many individuals across space and time is rare in animal studies. Some of our results are in line with previous work on artificial networks but also add new aspects to it. For instance, Evans et al. (2020) simulated a simple and conformity contagion on different social structures, and show that disease/information spread is generally faster (i.e. number of individuals infected/informed after a given time) for simple contagion compared to conformity contagion. Their findings show that the diffusion dynamics depend on the underlying social learning rule. Our study is in line with their findings in the sense that also our results suggest that diffusion dynamics depend on the considered learning rule. However, while Evans et al. (2020) investigated the ‘efficiency’ of spread (i.e. time until a certain number of individuals had been infected/informed), we specifically focus on the order in which individuals of different network positions acquired novel behaviour. Throughout the revised manuscript, we highlighted in more detail how our study is different and complementary to other studies (e.g. lines 58-69, 89-95, 134-142, 641-645; line numbers refer to the revised manuscript). Further, we believe that considering empirical work in addition to simulation work is crucial to test whether findings are the same or different between approaches. The reviewer’s next comment also highlights the importance of considering empirical networks because different species may provide different results. Simulation studies and more empirical studies both provide their advantages and disadvantages, and we believe that considering both approaches supplementary to each other will provide valuable information on how different contagions and sociality shape behavioural diffusion (we discuss this point in lines 586-602).

Evans, J. C., Silk, M. J., Boogert, N. J., and Hodgson, D. J. (2020). Infected or informed? Social structure and the simultaneous transmission of information and infectious disease. *Oikos*, *129*(9), 1271-1288.

2. The results depend on the characteristics of great tit networks. Although many different networks were used in the simulations, we can expect different results if networks of other species would be used. It would be helpful if the authors can provide some information about the structure of the empirical networks, and perhaps some predictions regarding their relevance to other species.

We have acted on this comment both in regards to quantification of great tit social structure and also in terms of generalising this to other study systems more broadly. Specifically, we have added histograms of the distribution of four global network metrics (i.e. network density, modularity, average edge strength, average path length) depicting the great tit social network structure (see Figure S2). Great tits form fission-fusion societies with groups of variable size and composition in which individuals join, leave, and re-join groups at frequent intervals. Fission-fusion societies are not only widespread among other bird species (see references within Silk et al. 2014) but frequently found in mammals (e.g. different species of primates (Amici et al. 2008), hyenas (Smith et al. 2007), dolphins (Lusseau et al. 2006)) and fish (e.g. guppies (Wilson et al. 2014), reef sharks (Papastamatiou et al. 2020)). Therefore, our observed social structures are likely meaningful for other species as well. However, many species live in highly stable groups (e.g. many primates) which may provide different results. Such species may for example express a lower variation in social connectivity (i.e. homogenous degree distribution) in which case an individual’s social network position may, under certain behavioural contagions, not have a strong impact on the probability of information acquisition (in contrast to social structures with heterogenous degree distributions such as expected in more gregarious species). Thus, our findings may differ to species living in highly stable social structures and it would be interesting for future work to test how different behavioural contagions spread on social networks of species with contrasting social structures (and the framework we use here would enable this). Yet, the common point remains that – across all systems – the type of contagion in play will shape which individuals (in regards to their network position) acquire and transmit this contagion. We added more information on the great tit social structure in lines 165172, 240-247 (and Figure S2) and discuss in more detail how our results may relate to other species in lines 583-602.

Aplin LM, Farine DR, Morand-Ferron J, Cockburn A, Thornton A, Sheldon BC. 2015. Experimentally induced innovations lead to persistent culture via conformity in wild birds. Nature. 518(7540):538–541.

Centola D. 2018. How behavior spreads: The science of complex contagions. Princeton University Press Princeton, NJ.

Danchin E, Nöbel S, Pocheville A, Dagaeff A-C, Demay L, Alphand M, Ranty-Roby S, Van Renssen L, Monier M, Gazagne E. 2018. Cultural flies: Conformist social learning in fruitflies predicts long-lasting mate-choice traditions. Science (80- ). 362(6418):1025–1030.

Firth JA, Albery GF, Beck KB, Jarić I, Spurgin LG, Sheldon BC, Hoppitt W. 2020. Analysing the Social Spread of Behaviour: Integrating Complex Contagions into Network Based Diffusions. arXiv Prepr arXiv201208925.

Hoppitt W, Laland KN. 2013. Social learning. Princeton University Press.

Kendal RL, Boogert NJ, Rendell L, Laland KN, Webster M, Jones PL. 2018. Social learning strategies: Bridge-building between fields. Trends Cogn Sci. 22(7):651–665.

Lachlan RF, Ratmann O, Nowicki S. 2018. Cultural conformity generates extremely stable traditions in bird song. Nat Commun. 9(1):1–9.

Rosenthal SB, Twomey CR, Hartnett AT, Wu HS, Couzin ID. 2015. Revealing the hidden networks of interaction in mobile animal groups allows prediction of complex behavioral contagion. Proc Natl Acad Sci. 112(15):4690–4695.

Van de Waal E, Borgeaud C, Whiten A. 2013. Potent social learning and conformity shape a wild primate’s foraging decisions. Science (80- ). 340(6131):483–485.

Whiten A. 2005. The second inheritance system of chimpanzees and humans. Nature. 437(7055):52–55.

3. As far as I could tell, the specific code and data for running these simulations were not shared (not just the package). This makes it impossible to replicate the results and assess the data structure.

We apologize for not providing the code and data earlier. Data and code to replicate all results are now available at https://osf.io/6jrhz/.

4. Learning rules in the wild were briefly mentioned in the introduction. Is there any evidence for the rules being used in wild animal populations, supporting the inclusion of these four rules?

Many different social learning rules are used within animal populations (Hoppitt and Laland 2013; Kendal et al. 2018), and we reference such work throughout our text. More specifically, several studies provide evidence for the conformity learning rule where individuals are disproportionately more likely to acquire behaviour when that behaviour is being performed by the majority of the population (Whiten 2005; Van de Waal et al. 2013; Aplin et al. 2015; Danchin et al. 2018; Lachlan et al. 2018). The proportional learning rule is conceptually just a special case of the conformity rule (i.e. just modifying the shape of the curve (Firth et al. 2020)); as such it is well in the scope of many animal species and only differentiates from the conformity rule in the assumption that the rate of social transmission is proportional (rather than disproportional). In contrast to studies on human behaviour (e.g. see references within Centola 2018), there is only little research on threshold-type learning in animals (but see Rosenthal et al. 2015). We added more information on the evidence for these learning rules in our revised introduction and provided a more in-depth exploration of previous theory/discussion about these rules beyond ecology (lines 55-58, 124-142). Finally, we believe that our work here and the encouragement of such approaches will increase the evidence for learning rules in the wild. Specifically, the current NBDA approach does not allow users to directly test for such learning rules, but these new empirical approaches (as applied here) will allow investigation (and evidence for) learning rules.

Aplin LM, Farine DR, Morand-Ferron J, Cockburn A, Thornton A, Sheldon BC. 2015. Experimentally induced innovations lead to persistent culture via conformity in wild birds. Nature. 518(7540):538–541.

Centola D. 2018. How behavior spreads: The science of complex contagions. Princeton University Press Princeton, NJ.

Danchin E, Nöbel S, Pocheville A, Dagaeff A-C, Demay L, Alphand M, Ranty-Roby S, Van Renssen L, Monier M, Gazagne E. 2018. Cultural flies: Conformist social learning in fruitflies predicts long-lasting mate-choice traditions. Science (80- ). 362(6418):1025–1030.

Firth JA, Albery GF, Beck KB, Jarić I, Spurgin LG, Sheldon BC, Hoppitt W. 2020. Analysing the Social Spread of Behaviour: Integrating Complex Contagions into Network Based Diffusions. arXiv Prepr arXiv201208925.

Hoppitt W, Laland KN. 2013. Social learning. Princeton University Press.

Kendal RL, Boogert NJ, Rendell L, Laland KN, Webster M, Jones PL. 2018. Social learning strategies: Bridge-building between fields. Trends Cogn Sci. 22(7):651–665.

Lachlan RF, Ratmann O, Nowicki S. 2018. Cultural conformity generates extremely stable traditions in bird song. Nat Commun. 9(1):1–9.

Rosenthal SB, Twomey CR, Hartnett AT, Wu HS, Couzin ID. 2015. Revealing the hidden networks of interaction in mobile animal groups allows prediction of complex behavioral contagion. Proc Natl Acad Sci. 112(15):4690–4695.

Van de Waal E, Borgeaud C, Whiten A. 2013. Potent social learning and conformity shape a wild primate’s foraging decisions. Science (80- ). 340(6131):483–485.

Whiten A. 2005. The second inheritance system of chimpanzees and humans. Nature. 437(7055):52–55.

5. Figure 2 and S2: I don't really understand why the order of acquisition is plotted as predicting the network traits (e.g., clustering coefficient). If there is any connection between the two it should be the other way around. To me it makes these figures confusing. Perhaps related, at least in the lower panels all individuals seem to have very similar trait values (e.g., weighted degree) -- how can this be explained?

Yes, we would expect that the network trait predicts the order of acquisition (OAC), as we discuss in the text in some depth. However, in these figures, we intended to show the OAC as a sequential event which is why we presented the OAC on the x-axis. Further, we only aimed at illustrating the correlations between the OAC and individual network metric and we did not perform analysis where one variable was predicting the other. Last, measures of the network metrics are scaled (centred and divided by the standard deviation) and cannot be as appropriately divided into separate ranks/groups such as the OAC. If we calculate a mean OAC (y-axis) for each value of network metric (x-axis), this results in mean OACs for various different measures of the network metrics and produces a figure that is difficult to read (see Author response image 1). Due to problems visualising our results that way, and because the other two reviewers did not raise concerns about Figure 2/S2, we decided to leave the OAC on the x-axis. The previous Figure 2/S2 showed averaged network metrics for each OAC. In the lower panels, averaged network metrics across the OAC all result around 0 because there is no apparent relationship between the OAC and network metrics across all network sizes and simulation runs when using the proportion and conformity learning rule.

**Author response image 1. sa2fig1:** Relationship between order of acquisition and weighted degree for the simple learning rule. Lines plot the average OAC for each value of weighted degree (scaled) and network size from the 100 simulations of each network. Colour represents network size with darker colour indicating smaller networks.

6. L. 94-100: Earlier, the authors mention that individuals may tend to learn from specific individuals such as relatives or with a given trait value. To me that sounds like a very likely mechanism, such as "learn only from your mother, or from your strongest affiliation". I would be very interested to see versions of such rules tested here.

We agree that many different rules (other than the ones used here) would be very interesting to test. In this particular case however, great tits are relatively short-lived with a mean life span of about 1.9 years (Bulmer and Perrins 1973). Short life spans result in high annual population turnover of about 50% which results in no prominent kin structure and less than 1.5% of winter social connections are between first-order relatives (Firth and Sheldon 2016). We added this information in lines 165-170. Therefore, a learning rule based on ‘only learn from your mother’ is rather unlikely in tits, given the lack of kin structure in this population. However, such a rule might be highly relevant for species with more prominent kin-structures, e.g. where mother-offspring bonds are stronger than bonds with other group members. We highlight in our revised discussion that testing such a rule in future work would be very interesting (lines 703-707). A rule based on ‘learn from your strongest affiliation’ may result in behaviours not spreading rapidly because if the strongest affiliate of individual A is individual B, it is very likely that the strongest affiliate of individual B is individual A. For instance, in great tits future social pairs often have the strongest association strength. Therefore, for such a behaviour to spread across a population would require a substantial amount of asocial learning events. Nevertheless, we believe that our work here generally encourages readers to consider which social learning rules may be operating in their system, and provides the conceptual framework to test a number of rules.

Bulmer MG, Perrins CM. 1973. Mortality in the great tit Parus major. Ibis (Lond 1859). 115(2):277–281.

Firth JA, Sheldon BC. 2016. Social carry-over effects underpin trans-seasonally linked structure in a wild bird population. Ecol Lett. 19(11):1324–1332.

L. 90: Did you mean predator avoidance?

We refer here to studies providing evidence for social information transmission on the avoidance of aposematic prey by predators (blue tits and great tits).

L. 152: The introduction mentioned weekly networks but here weekends are reported. How much time does each network include?

We apologies for the unclarity. The ‘weekly’ networks only represent a weekly snapshot of data collected over a weekend (i.e. this is the weekly sampling) and thus represent two-day networks. We clarify now in line 118-119.

Figure 1: This figure is referenced only from the methods section. It raises a question: Why were the networks generated from only one location each time? I assume that two birds may have also interacted in adjacent locations. Perhaps some data on the overlap across adjacent locations can help.

In agreement with the editors, the methods section now proceeds the Results section in our revised manuscript and therefore Figure 1 is referenced earlier. We have several different reasons for why we created social networks for each feeder location separately, and understanding the spread of behaviours within local populations was a key aspect of this manuscript (rather than between local populations). More specifically: First, we generated local social networks to remove any spatial effects influencing the probability of social behavioural acquisition. For instance, within a subpopulation where a new behaviour emerges, individuals with high connectivity may be faster in adopting the behaviour. However, if examined on the population-level, such a relationship may be obscured by spatial effects, because an individual’s probability of behavioural adoption will be considerably predicted by its spatial proximity to the location of behavioural emergence, regardless of its overall social network position within the whole population. Further, focusing on one feeder location provides a comparable spatial unit and does not require to draw arbitrary spatial boundaries across the study site. Second, in our study system birds on average only visit 1.3 feeder locations on a given weekend (min=1, max=10, sd=0.7) and from 21036 occasions where individuals were recorded on a given weekend, individuals had visited only one location in 14888 occasions (71%). Therefore, in the majority of cases, individuals only visited one feeder on a weekend. We added this information in lines 364-373. Last, we aimed at testing behavioural diffusion on a large variation of different social networks which we acquired by calculating weekly and local social networks. Creating only weekly networks across the whole population would have resulted in only 39 social networks (compared to 1343 weekly, local networks). We added a justification of why we calculated local social networks in lines 199-207.

L. 206-207: This statement does not seem to be supported by Figure 3, in which the correlation coefficients for weighted degree and weighted betweenness are actually lower in the simple learning rule.

We meant to say that the simple learning rule created overall the strongest positive and negative correlations. We clarify now in line 409-411.

L. 248…: I was surprised to see here, in the last part of the results, an explanation of the simulation procedure, which belongs either to the methods section, or to the beginning of the results.

In agreement with the editors, we re-structured our manuscript so that the methods section proceeds the Results section. We therefore, removed the explanation of the simulation procedure in the Results section and give a more detailed explanation in the methods (see section on ‘Simulations’) which will hopefully increase clarity.

L. 470: foal should be focal

Changed in line 230.

Reviewer #2 (Recommendations for the authors):Overall, I found the article interesting and feel that it provides an interesting examination of the consequences of how individuals learn on how behavioural contagions spread through animal societies. Previous studies have not applied these types of model to empirically-derived animal network data or studied as many forms of behavioural contagion simultaneously. Consequently, while (in the context of broader network science research) the individual results here are not new, it is valuable to see them presented together and in a way that is accessible to behavioural ecologists.I felt the article was well-framed in the introduction, and the results were largely presented clearly and intuitively. In particular, I thought that Figures 2 and 5 were an excellent way to clearly show some of the main results of the study. However, there was insufficient information provided about the models or the experimental design of the modelling component prior to the results, which made interpretation of model results harder than it should be and at times made the results hard to follow when intertwined with methodological information.

Thank you for the encouraging comments here, and for the suggestions. In agreement with the editors, we re-structured our manuscript so that the methods section proceeds the Results section which will hopefully increase clarity. Further, we added a more detailed description of our modelling procedure, including all necessary equations (see methods sections on ‘Simulations’) and provide all R code and data to replicate our results.

I have some concerns about the modelling approach used, although perhaps some of them are related to the rather limited information provided on the simulation approach (I would really encourage the authors to provide clearer methods that incorporate functions for the learning rules and a step-by-step description of the simulation algorithm). It is not clear to me whether acquisition is deterministic based on likelihoods or probabilistic, and the step explaining how the likelihood of information being acquired socially or asocially is particularly unclear. This plays in to some concerns about the appropriateness of a strictly order-based algorithm -- it seems a rather artificial choice based on an existing statistical model rather than clearly biologically justified. Given the shape of the adoption curves for different contagions will differ, and that there is a probability of asocial component, avoiding a time-based approach seems like it could potentially have different consequences for different social learning rules. I can potentially see an argument that individuals would never truly acquire information at the "same" time, but this perhaps raises the question as to: (a) how meaningful changes in behaviour might or might not be to the subsequent individual to acquire the behaviour; and (b) how biologically meaningful differences in adoption time are (outside of very particular biological contexts such as during a predation event).

Thank you for this comment. Firstly, in response to the aspects of clarity around our simulation description, we have reworded various parts of this description (as well as placing it before the Results), and we added a more detailed step-by-step explanation of our simulation algorithm, including all relevant equations in our methods section ‘Simulations’ which will hopefully improve clarity. Secondly, in regards to order of acquisition – We were specifically interested in the relationship between individual sociality and the order of acquisition and not necessarily in the time it takes to inform a given proportion of the population (which is the focus of many other studies, e.g. Evans et al. 2021) or in the exact time when individuals adopt a behaviour. This is due to two things:

(1) even though our networks are empirical, the diffusions upon them are simulated and as such the order-based approaches make less assumptions (and require less parameters) than the time-based approaches. (2) Within natural populations with empirical diffusions the order-based approach is one of the most used and most generalisable approach for considering the spread of information and behaviours (albeit within the context of simple NBDA), and as such this work is a close match to those empirical pieces. We agree that the ‘order of acquisition’ is somewhat artificial and that in natural settings, individuals will adopt a novel behaviour during different times with some individuals adopting the behaviour almost simultaneously while others adopt behaviours much sooner than others. Nevertheless, in reality, our observation of natural systems often matches the order of acquisition assumptions more closely, as it is uncommon to know the exact timing of acquisition for each individual, but rather to have ‘snapshots’ of which individuals are likely to have acquired in which order. We also believe that in simulation studies that examine the timing of acquisition, the times at which individuals adopt a novel behaviour will be highly dependent on the model’s parameter settings and thus are also artificial measures. Therefore, meaningful measures for between individual variation in the timing of acquisition can only be acquired empirically in systems with complete observations. How biologically meaningful differences in the order of acquisition are will ultimately depend on the behaviour and context, and the actual observed variation in adoption times. For instance, we find that under simple contagions, individuals with a higher weighted degree adopt novel behaviours sooner. Under natural settings this could mean that individuals with a higher weighted degree acquire information about e.g. a novel food source sooner than individuals with a lower weighted degree. Whether this difference in the order of behavioural adoption is biologically meaningful will depend on the variation in time of acquisition. For instance, if the whole population acquires the same information within a few minutes than the benefit of individuals with a higher weighted degree finding the novel food source sooner will be smaller than if information spread takes several days or weeks. If the latter, individuals with a lower weighted degree may miss out on a potentially highly profitable food source for several days. We mention these points in our revised discussion (lines 626-630; line numbers refer to the revised manuscript).

Evans JC, Hodgson DJ, Boogert NJ, Silk MJ. 2021. Group size and modularity interact to shape the spread of infection and information through animal societies. Behav Ecol Sociobiol. 75(12):1–14.

Another concern of the work presented here is that there are minimal checks of sensitivity to model parameterization (aside from changing the threshold number of connections in the threshold model). Parameter selection for the different forms of social learning is fairly weakly justified and not well explained so it is not clear what effect this might have on the generalizability of the results. For example, previous modeling work has suggested that the difference (in speed) in how simple contagions and contagions based on conformist learning spread through some types of networks depends on how easy the contagion can spread. This may not impact results related to the order of acquisition, but it is hard to tell if this is likely to be the case, especially given there is also an asocial learning component that has an impact on the adoption of behaviours.

We apologise for this shortcoming. In our revised manuscript, we repeated the simulations with different parameter combinations for each of the different simulations (see section on ‘Simulations’) and provide the extensive additional results in the Supplementary Material. Specifically, we altered the ‘s’ parameter which denotes the social transmission rate. We tested ‘s’ values of 1, 5, 10. We further altered the frequency dependence parameter ‘f’ in the conformity model and the threshold location parameter ‘a’ in the threshold model (f/a: 3, 5, 7). In the proportion model, the frequency dependence parameter is per definition 1. If it is >1 it breaks down to the conformity model. In the main text, we present results on the parameters s=5, f=5, a=5, and present results on smaller and larger parameters in the supplementary material (Figure S6-S15). Overall, altering these parameters did not change our general findings and conclusions. However, for instance, the probability of social spread increased with increasing social transmission rate (Figure 14) and larger threshold locations (i.e. larger values for a) decreased the probability of social spread (Figure 15). In addition, we now provide all data and R code with which results can be replicated and readers can test different parameters themselves.

I also found it interesting that the networks were weighted using simple ratio indices, and the reasoning for this wasn't clear. There are some important assumptions hidden in this choice that could potentially impact the results of the study related to what these indices relate to and how individuals learn. An (intentionally) naïve starting point (in my mind at least) would be that the likelihood of social learning depends on the number of times individuals associate at a food resource rather than a proportion of times that they occur together (e.g., individuals that occur together 4 out of 10 observations may well have more opportunities to learn than 1 out of 2 times). This is clearly a very direct interpretation of the simple ratio indices which ignores the potential for associations elsewhere and their "representative value". It is also based on the assumption that number of associations is the factor that drives social learning (as opposed to, e.g., associations, within a set amount of time, strength of social bond, etc.). However, it would be good to more clearly set out the reasoning for this choice and perhaps test or discuss the sensitivity to different assumptions here.

The Simple Ratio Index is widely used in animal social network studies to give a good proxy of association strength between two individuals (and network position among individuals) given the factors that play into these systems, such as differences in observation (Hoppitt and Farine 2018). Much previous research has justified this as a useful measure for estimating the ‘true’ social network, as well as for relating this inferred network to the network at other contexts, and also in other social processes. So, yes, intuitively in a perfectly observed system an individual in the former case described by the reviewer (4 out of 10) may have more opportunities to learn at the feeder compared to the latter case (1 out of 2) at that particular feeder. If one would have equal numbers of observations for all individuals in the group or population, measuring the association strength based on simply the number of times two individuals were seen together might also be appropriate. However, in our study and in most animal network studies, the number of observations differ between individuals – and also the researcher may be interested in not just the associations observed in that one particular sampling context but also in contexts outside of that. In these common cases, the social association between two individuals is better represented as a ratio (Farine and Whitehead 2015). Therefore, we intended to use the SRI as a representative value for the general social relationships between individuals, not necessarily to just reflect opportunities for learning at the feeder. For instance, learning could also happen away from the feeder and we thus wanted to get a generalized measure of ‘association strength’. Last, the SRI was used as a measure to infer weighted social networks in various studies on great tits showing that the weighted edges between individuals are meaningful in predicting a whole range of population processes (e.g. breeding settlement (Firth and Sheldon 2016), mating (Culina et al. 2015), and information transmission (Aplin et al. 2015)). We added a better justification for why we choose the SRI in lines 216-223.

Aplin LM, Farine DR, Morand-Ferron J, Cockburn A, Thornton A, Sheldon BC. 2015. Experimentally induced innovations lead to persistent culture via conformity in wild birds. Nature. 518(7540):538–541.

Culina A, Hinde CA, Sheldon BC. 2015. Carry-over effects of the social environment on future divorce probability in a wild bird population. Proc R Soc B Biol Sci. 282(1817):20150920.

Farine DR, Whitehead H. 2015. Constructing, conducting and interpreting animal social network analysis. J Anim Ecol. 84(5):1144–1163.

Firth JA, Sheldon BC. 2016. Social carry-over effects underpin trans-seasonally linked structure in a wild bird population. Ecol Lett. 19(11):1324–1332.

Hoppitt, W. J., and Farine, D. R. (2018). Association indices for quantifying social relationships: how to deal with missing observations of individuals or groups. Animal Behaviour. 136:227-238.

Away from the research itself, one frustration when reading this paper is that it does a fairly poor job of placing itself in the context of the wider literature. First, while it does a good job of citing the relevant studies that conduct similar modelling work in animal societies, there is relatively little effort to engage with the findings of these studies in the introduction or discussion of the paper. There is a real opportunity here to unpack the results of this study in relation to similar and contrasting findings from other papers that is missed here. Different papers have focused on different aspects of how social structure and connections influence contagions in animal societies and by linking better with some of these papers it could perhaps address how the findings of this study might generalize (e.g., to different social structures, considering different "transmissibilities" of contagions, etc.). Second, there is little effort made to acknowledge or consider the large number of modelling studies that address similar questions in the broader network science literature. While the network here is empirically derived, from that point on this study is purely computational and there are studies that have addressed very similar or overlapping questions elsewhere in this literature (e.g., how the number of connections influences speed of acquisition for different forms of contagion).

We apologise for this shortcoming. In our revised manuscript, we cite and engaged more with the findings of other studies and implement more studies from fields of sociology and network science in the revised introduction (e.g. 79-82, 89-96) and discussion (e.g. 572-575, 640-645). We would also be happy to include any specific works that the reviewer may be referring to here if they want to include those too.

Related to this point, it would be nice to see a more nuanced discussion about the strengths and weaknesses of computational research that either applies simulation models to a single empirical case study vs. that the applies similar computational models to more generalized network structures. While there was a point when these types of model were applied to very generic network structures (random, small-world, etc.) and to an extent still are in network science (where the research aims are somewhat different), more frequently now studies that use simulated network structures do so with express biological questions in mind and design simulated networks accordingly. Taking this approach is a powerful way of tackling specific questions and/or generating a range of generalizable structures. Equally applying these types of models to particular empirical case studies is very valuable in its own right for different reasons. Related to the previous point, I think it would be great to make the most of their complementary strengths to better integrate the lessons learned from these different approaches.

Yes, we agree that both empirical and simulation studies can provide valuable results on their own, and that considering both complementary to each other will improve our understanding of how behaviours spread and how individual sociality relates to the probability of behaviour acquisition. We highlight in lines 718-729.

This recently published paper is perhaps relevant/useful:https://royalsocietypublishing.org/doi/full/10.1098/rspb.2022.1001

Thank you for forwarding this paper. This is indeed a very relevant study and we included it in our revised manuscript (e.g. line 741).

L20: Given the weak correlations illustrated in Figure 3, it feels slightly misleading to describe this as a "strong" relationship.

We agree and removed the word ‘strong’ in line 20.

L22-23 (and elsewhere): Discussion of this idea throughout the paper doesn't acknowledge previous work showing this outside ecology. This review contains more links to studies in network science as a useful resource https://onlinelibrary.wiley.com/doi/full/10.1111/oik.07148?saml_referrer. For example, this paper https://journals.plos.org/plosone/article?id=10.1371/journal.pone.0020207 tackles how conformist social learning leads to this pattern.

Thank you for drawing our attention to these relevant papers. We acknowledge more previous work in sociology and network science in our revised introduction and discussion (e.g. 79-82, 134-142, 572-575, 639-645).

L48: "requiring exposure to multiple sources" isn't necessarily a difference between infection and behaviour spread in networks (e.g., see literature on dose-response curves).

Yes, we agree that this is not necessarily a difference between infection and behaviour spread. In this sentence we meant to contrast this statement with the frequent assumption of social learning, i.e. that the extent (i.e. number and duration) of social contacts to knowledgeable others predicts the likelihood of adoption, and not contrasting information transmission with disease transmission (see line 49).

L76-78: Evans et al. (2021) also consider a simple contagion and conformist learning and explore the potential implications of considering the simple contagion as being something other than the spread of infection in the discussion.

We included the relevant reference throughout the revised manuscript (e.g. line 60).

L83-85: While I think a really interesting aspect of this study is its exploration of the role of network size, I am not sure how well this criticism works (in its current form) given that even big networks can capture interaction patterns at small spatial scales, and that how network structure depends on the temporal scale will be highly study-system dependent.

We agree that also large networks can capture interaction patterns at small spatial scales.

Generating local social networks was done for numerous reasons (see response to reviewer 1 above) including to remove spatial effects between local populations from our analysis. When examining the relationship between individual sociality and the probability of adoption, generating a social network from the whole population would add considerable spatial effects. For instance, within a sub-population where a new behaviour emerges, individuals with high connectivity may be faster in adopting the behaviour. However, if examined on the population-level, such a relationship may be obscured by spatial effects, because an individual’s probability of behavioural adoption will be considerably predicted by its’ spatial proximity to the location of behavioural emergence. Further, we aimed at testing behavioural diffusion on a large variation of different social networks which we acquired by calculating weekly and local social networks. Creating only weekly networks across the whole population would have resulted in only 39 social networks (compared to 1343 weekly, local networks). We added a justification of why we calculated local social networks in lines 199-207 and re-wrote the section in the introduction accordingly (lines 101-103).

We agree that to which extent social structure depends on the temporal scale will be very study system dependent. Great tits forage in fission-fusion flocks during the winter and thus their group size and composition (and thus their social connections) frequently changes across the winter. Contrary, in more stable social structures (such as found in many primate species), social connections between individuals are likely very stable across time and are subject to little changes. We clarify in lines 101-112.

L96-100: Given the methods come last, these rules need to be more accurately described here to help with the interpretation of the results. It would also be good to provide a more in-depth exploration of previous theory/discussion about these rules beyond ecology.

In agreement with the editors, we re-structured our manuscript so that the methods section proceeds the Results section. We hope this improves the clarity of our methods. In addition, we included a more in-depth discussion of previous theory and evidence for the four learning rules in lines 124-142.

L112-113: Low clustering coefficient will not always indicate a less sociable individual, this will depend considerably on the broader structure of the network. It would seem to in this study, but some greater context would be helpful here. In the results it would be helpful to quickly describe the correlation between the centrality measures used in the studied networks.

We agree that a low clustering coefficient will not always indicate a less sociable individual and rewrote accordingly across the whole revised manuscript. The three individual network metrics are moderately correlated, with weighted clustering coefficient being negatively correlated to weighted degree and weighted betweenness, and weighted degree and weighted betweenness were positively correlated (see Table S1). In addition, we provide example networks with individual great tits colour-coded based on their different weighted network metrics (Figure S1) and calculated four global network measures (network density, average path length, average edge weight, modularity) to provide a general overview of the weekly, local great tit social structures (Figure S2, details on how these metrics were calculated can be found in the figure legend).

Figure 1: Would be useful (and take up no extra space) to give the thresholds used here.

We have not used any thresholding in Figure 1. Figure 1 is purely illustrative showing the simulation processes. We now ensure this is clear from the figure caption. We speculate that the reviewer was referring to Figure 2 and we added all parameters used (including the threshold parameters) to produce the respective results in the figure legends of the revised Figure 2 and 5.

L151-154: It would be good to provide more clear information on networks per feeder site, network appearances per bird, etc., and an indication of what timespan were data included from. Also perhaps valuable to point out that the minimum of 10 (and rest of these descriptive stats) applies after some networks were excluded.

We provide more detailed information on these points in lines 176-179 and 364-373. For each location, we included on average 21.7 networks into the analysis (min=1, max=39, sd=11.9) and each individual was part of on average 16.5 networks (min=1, max=88, sd=13.2). In each of the three winters (2011-12, 2012-2013, 2013-2014) the feeders were in place from December to February and collected data on the bird visits from pre-dawn Saturday morning until after dusk on Sunday evening. Feeder locations were consistent across the three years.

Thank you for pointing this out. We clarify that the minimum of 10 only applies after our data exclusion (lines 367-368).

Figure 2: One thing I found particularly interesting in these results is that for simple, threshold and to some extent conformity there appears to be a stronger pattern of being the most peripheral is costly rather than being the most central is beneficial. Clearly, this may depend on what the information/behaviour that is spreading is related to, but it's a neat result and perhaps worthy of some discussion for what it means for social ecology/evolution in this system.

Thank you for pointing this out. This is indeed an interesting take on this result which may have important consequences for the associated costs and benefits of individual sociality. We discuss the view of this finding in lines 611-626.

L210: Is this now meant to refer to Figure 3?

We re-wrote this whole section in lines 433-457.

Figure 3/Figure 4: Even the correlations different from zero are (predominantly) small. Later in the paper it might be interesting to discuss how biologically meaningful they are in networks of different sizes in this system. One thing apparent in these results is that there is a lot of noise (presumably) related to the seeded individual. It may be worth using this to highlight that in small networks who you know is just as/more important than how many you know -- even if it is just as a suggestion for further research.

Thank you for pointing this out. Yes, correlation coefficients are generally small which might be due to multiple reasons. As noted in our manuscript (lines 411-416), this may result from general nonlinear relationships between network metrics and order of acquisition in real network structures like this. For instance, for the threshold model, the average network metric is close to 0 at first until it slightly increases and then decreases again for late adoptions (Figure 2) which may cause low correlation coefficients. Indeed, we also speculate that the starting position of the seeded behaviour (i.e. the network position of the demonstrator) and likely the underlying general social network structure (e.g. network density, modularity) cause a large variation in diffusion pathways, and thus correlation coefficients. We discuss in more detail the biological relevance of our findings in lines 626-630 and highlight that some of these aspects such as exploring the relevance of the starting position in shaping diffusion pathways would be very interesting for future work (lines 624-626).

Figure 4: I found distinguishing between the two blues here pretty difficult. Another colour scheme may be clearer or a different way of presenting the data given how dense the point clouds are.

We chose a different colour scheme across all figures and the revised Figure 4 shows now each learning rule in a separate plot.

L219-231: It would be good to make this aspect of the experimental design clearer earlier. I also found these results were written less clearly than other parts -- some rewriting might be helpful.

We re-wrote this section and hope it improved in clarity (lines 375-457). Further, we included information on testing different parameters already in the methods section (lines 324-330).

L248-255: Some indication of how the model broadly works such as this should ideally come at the end of the methods to help with interpreting the results in this methods-last format.

We restructured as suggested. Further, in agreement with the editors, the methods section proceeds the Results section in the revised manuscript. We hope this improves the clarity of our methods and results.

L270-274: Would perhaps be helpful at the end of the introduction to set up the model or in the methods?

We included this information in the methods section on ‘Simulations’.

L362-364: Is there not a little more nuance here given that there won't necessarily be a single learning rule for each behaviour so it suggests perhaps that the importance of different learning rules varies as a behaviour spreads.

Yes, these findings suggest that the importance of the social learning rule may be most prominent during the initial stages of spread. We made this clearer in line 655. Further, we agree that one limitation of our models is that each individual adopts a seeded behaviour under the same ‘learning rule’. However, individuals will likely differ in the extent of social information use. For future work, it would be very interesting to examine within and between individual variation in different learning rules. We discuss this limitation in lines 690-703.

L378-380: Similarly to the previous point, an appropriately parameterized (dynamic) network for a large population can capture interactions at very fine spatial and temporal scale -- this doesn't seem so much a point directly related to network size. One aspect of network size that perhaps becomes interesting is that the importance of "who" you know versus how many you know changes in importance in different size networks.

Yes, we agree that also networks of a whole population can capture associations on a fine spatiotemporal scale. As mentioned above, one of our main reasons for generating local social networks was to remove spatial structure per se from our analysis, and consider social transmission within local populations. Otherwise, when examining the relationship between individual sociality and the probability of adoption, generating a social network from the whole population would add considerable spatial noise since spatial proximity to the source would impact an individuals’ probability of social learning. In addition, we wanted to generate a large variety of social networks differing in structure and size (see lines 189-207). We agree that to which extent social structure depends on the temporal scale will be study-system dependent. Great tits forage in fission-fusion flocks during the winter and thus their group size/composition, and thus their social connections, frequently changes across the winter. For other species, in e.g. highly stable social groups this may be different. We re-wrote to clarify in lines 103-112. Related to this point, we now highlight the importance of considering dynamic versus static networks (686). Yes, we agree and discuss the potential difference in the importance of ‘who’ you know versus ‘how many’ you know in relation to network size in lines 671-676.

L383: Is there a reason for using network "shape" here rather than more established terms like topology or structure?

Rephrased to ‘network structure’ in line 689.

L387-405: Good to see a sensible consideration of model limitations.

Thank you.

L449-453: Is there empirical data to support this for this study system given it has been the subject of previous experimental work? Justifying with previous empirical work would strengthen this point considerably.

In Aplin et al. (2015a, b), authors report that the population-level bias for an introduced technique (i.e. which direction to push open a door to access food at a feeder) increased daily. Further, by measuring the proportion of individuals that were observed performing each behavioural variant (push left or right) in the social group that preceded a naïve bird’s first successful solution, Aplin et al. (2015b) show that individuals were disproportionately likely to copy the behavioural variant of the majority of individuals. This finding suggests that the social connections on a very small temporal scale (e.g. the social group preceding a focal individual’s first solve) are important in determining behavioural adoption. We added a more detailed description of our reasoning in lines 194-199.

Aplin LM, Farine DR, Morand-Ferron J, Cockburn A, Thornton A, Sheldon BC. 2015a. Experimentally induced innovations lead to persistent culture via conformity in wild birds. Nature. 518(7540):538–541.

Aplin LM, Farine DR, Morand-Ferron J, Cockburn A, Thornton A, Sheldon BC. 2015b. Counting conformity: evaluating the units of information in frequency-dependent social learning. Anim Behav. 110:e5–e8.

L449-453: What biases do you introduce to the networks by only including interactions at a single feeder? How many individuals use multiple feeders?

In this work, we aimed to consider the transmission of behaviours within local populations (rather than among local populations). On a given weekend, birds on average only visit 1.3 feeder locations (min=1, max=10) and from 21036 occasions where individuals were recorded on a given weekend, individuals had visited only one location in 14888 occasions (71%). We include this information now in the Results section (lines 364-373). While the majority of birds only visited one feeder, some birds did visit more feeders and thus the local social network position will not capture all of an individual’s social connections at a given weekend. Therefore, the social network position of an individual inferred at one location may not be representative of its’ inferred network position at another location or across the whole population. For instance, at location X, individual A may have few connections compared to other individuals at location X (because it may have visited location X only few times). Whereas at location Y, individual A might have many connections compared to other individuals at location Y. However, in our study we were specifically interested in capturing the local diffusion pathways. In this case, we would expect that if a novel behaviour emerges at location X, individual A may have a lower probability to socially adopt the behaviour compared to others, whereas at location Y, the situation may be reversed. Therefore, we believe that even though some of the local individual social network metrics may not be representative of an individual’s overall social network position it will not change our general findings and conclusions. Further, we expect that the overall social network structure is dependent on the spatial scale over which associations are considered. We speculate that networks inferred at only one location are for instance denser and less fragmented compared to networks generated across the whole population.

L470: I am not convinced the use of clique (in the strict network analysis definition) is correct here.

We re-wrote in lines 229-232.

L505: I would suggest making it clear earlier in this description that simulations were repeated multiple times in each network.

The methods section proceeds the Results section now and thus this information comes much earlier in the main text (lines 321).

L520: Is this number correct given different thresholds were used too for further simulation runs?

Thank you for pointing this out. Indeed, these numbers did not include the simulations carried out when using different threshold parameters. They only inferred to simulations with one threshold parameter (the one presented across all Figures in the main text). We clarify now in line 327.

L530-53: Would be good to provide confirmatory information on model goodness of fit checks and clarify how statistical inference was done (presumably from the full/fitted model?).

We included information on statistical inference in lines 346-347 and models can be replicated with the data and R code provided at https://osf.io/6jrhz/.

I hope my comments help improve the manuscript.

We thank the reviewer for the very constructive comments which greatly helped to improve our manuscript.

Reviewer #3 (Recommendations for the authors):This study is a timely theoretical exploration of how variation in transmission rules interacts with variation in social phenotype to influence diffusion dynamics. The authors predict that the likelihood of an individual adopting novel behaviors should depend on the learning rule, as well as the individual's sociality. The authors explore the spread of behavior under 4 different learning rules: a simple adoption rule, a threshold rule, a proportion rule and a conformity adoption rule. They quantify the sociality of individuals by calculating their weighted degree, clustering coefficient and betweenness.The authors find that under simple and threshold rules, high degree, high betweenness, and low clustered individuals acquired the seeded behavior earlier in simulations. Under proportional and conformity rules, there was no strong relationship between social phenotype and order of acquisition. The authors find that network size predicts the magnitude and direction of the correlation between social phenotype and order of acquisition and that this relationship also depends on the learning rules.Strengths1. Overall the paper is well written, and the motivation for using computational simulations is well warranted to explore this question.2. The topic is timely, and tackles an important theoretical question of how variation in learning rules might interact with social phenotype to influence cultural diffusions. This is a difficult topic to address but is critical for improving our understanding of how diffusions might differ between populations.3. The authors construct simulations using real social networks of great tits, rather than artificially generated networks, which is a rarity, and thus of great value, in the SL modeling literature. These networks are hard earned -- taken at a relatively fine temporal scale from weekly sampling.4. The authors provide a thorough discussion of the implications of their results.

Thank you very much for these positive comments. We were particularly encouraged reading the reviewer’s recognition that “…These networks are hard earned…”.

Weaknesses1. One main weaknesses of the paper is the lack of details given about the transmission model. The authors do not provide equations, descriptions of parameters, a detailed schedule of the model. The descriptions they do provide are spread throughout the manuscript, making it more difficult to assess. NBDA is definitely an appropriate model for the question they want to answer, but it seems like the authors have altered some features of the model (e.g., only 1 individual can learn per timestep, 1 individual must learn per timestep). This lack of clarity makes it harder to assess the results they present. Further, from their description, it appears that they have allowed for asocial learning, which adds unnecessary noise to a study that is focused on social transmission.

We apologise for this shortcoming. In agreement with the editors the methods section now proceeds the Results section which we hope will improve clarity. In addition, we added a more detailed description of our modelling procedure, including all relevant equations (see method section on ‘Simulations’) and test different parameters (lines 325-330; line numbers refer to the revised manuscript). Further, we provide now all data and code so that readers can replicate our simulations at https://osf.io/6jrhz/.

In regards to the point about altered features in the NBDA – We were specifically interested in the relationship between individual sociality and the order of acquisition which is why only one individual could adopt the behaviour at a time, and not necessarily in the time it takes to inform a given proportion of the population (which is the focus of many other studies, e.g. Evans et al. 2021) or in the exact time when individuals adopt a behaviour. The ‘order of acquisition’ is of course somewhat artificial and in natural settings, individuals will adopt a novel behaviour during different times with some individuals adopting the behaviour almost simultaneously while others adopting behaviours much sooner than others (see also response to reviewer 2 above). Nevertheless, in reality, our observation of natural systems often matches the order of acquisition assumptions more closely, as it is uncommon to know the exact timing of acquisition for each individual, but rather to have ‘snapshots’ of which individuals are likely to have acquired in which order. We also believe that in simulation studies that examine the timing of acquisition, the times at which individuals adopt a

novel behaviour will be highly dependent on the model’s parameter settings and thus are also artificial measures. Our reasoning for ‘a new individual learning at each timestep’ was because we wanted to examine the relationship between individual sociality and the order of acquisition across the whole network. Without such a rule (and without allowing for asocial learning, see below), behaviours would not socially spread far (or at all) under certain circumstances (e.g. dependent on the position of the demonstrator or the underlying learning rule).

Finally, in regards to the point about asocial learning: ‘Asocial learning’ here is simply used to describe the acquisition of the behaviour by an individual that is not observed to take place due to the links within the observed social network. As such, some ‘asocial learning’ is necessary for the analysis to work at all, as without it there would be zero individuals with the behaviour to begin and this would persist throughout the simulation. Therefore, ‘allowing’ for asocial learning is a necessity, and then the question becomes how much asocial learning should be allowed for. We believe that the reviewer here is not arguing for not allowing asocial learning, but for only allowing for the very minimum amount of asocial learning (i.e. 1 asocial learning event per simulation per local population), and then banning asocial learning from that point onwards. However, we do not know of an animal system were such a scenario is likely to take place in natural settings, nor do we know of any previous empirical work that has dismissed the possibility of asocial learning entirely after the first asocial learning event. Therefore, we have opted to match what we believe would be relevant to the systems considered here, and continue allowing asocial learning after the first asocial learning event (indeed, this is also a major conceptual aspect of NBDA approaches generally too). Nevertheless, as we now provide all the data and code to replicate our simulations, the more interested readers are free to modify this to their desired asocial learning levels. However, we would add caution against considering zero possibility of asocial learning after the first asocial learning event, especially as ‘asocial learning’ not only captures the chance of an individual acquiring the behaviour themselves, but also captures any events were the individual acquires the behaviour socially but from outside of the observed network. Scenarios where we could totally write off both of these possibilities in animal social systems are very rare.

2. Another main weakness is that the authors do not use a sensitivity analysis, and thus it is difficult to assess the relative effects of each network metric, as they are not necessarily independent of one another. For example, degree and clustering can be correlated simply as a result of how clustering is calculated. This is the downside of using real networks, as without synthetic data, there may be insufficient data to perform a sensitivity analysis. Further, the authors do not present an assessment of variation in their results, instead showing mean values within network size as evidence of their claims.

Yes, a downside of using empirical networks is that it is difficult to perform a sensitivity analysis to assess the effects of each network metric on its own. However, in our study, we did not intend to quantify the relative effect of each network metric separately and also make no claims of that sort (e.g. by suggesting that a high weighted degree increases the likelihood of behavioural adoption more than a high weighted betweenness). We aimed at choosing three metrics that are commonly used in animal social network studies, and that also describe different properties of an individual’s network position and therefore roughly represent less and more social individuals. In our social networks, weighted degree and weighted betweenness are positively correlated, and weighted degree and weighted clustering, and weighted betweenness and weighted clustering are negative correlated (Table S1). In our manuscript, we often described individuals with a lower clustering, and high degree and betweenness as more social individuals. We agree that this is not appropriate and have been more careful in our wording in the revised manuscript. In addition to the correlation coefficients, we added a supplementary figure (Figure S1) illustrating a few social network examples colour-coding the range of network metrics which will hopefully help illustrate the overall correlation in network metrics. In our revised manuscript, we discuss some of the advantages and disadvantages of empirical and pure computational studies and highlight here that one key strength of computational studies is the ability to examine the relative contribution of parameters considered (e.g. by sensitivity analysis, lines 721-731).

Showing confidence intervals in Figure 2 and 5 would have not benefited visualization because lines would have highly overlapped. However, in our revised figures, we binned network size into different groups and now present, next to the mean, also the 95% confidence intervals (e.g. Figure 2, 5) and the revised Figure 3 and Figure 4 show the same measures of variation as before (violin and boxplots showing 25%, median and 75% of data in Figure 3, and 95% confidence intervals in Figure 4).

3. Related to the interpretation of sociality, there is opportunity to increase clarity. The authors describe more social individuals as having a high degree, high betweenness, and low clustering, and less social individuals as low degree, low betweenness and high clustering. One could also imagine a bird who has high degree, low betweenness, and high clustering, being at the center of their group, but rarely going between groups. It seems harder to argue that this bird is less social than a bird with high degree and high betweenness but low clustering. The manuscript would benefit from a careful description of how different combinations of these social metrics could be interpreted.

We agree that our wording was previously unclear in this sense and we have chosen a more careful description throughout the revised manuscript. In our social networks, weighted degree and weighted betweenness are positively correlated, and weighted degree and weighted clustering, and weighted betweenness and weighted clustering are negative correlated (Table S1). In addition, we added a supplementary figure (Figure S1) illustrating a few social network examples colour-coding the range of network metrics which will hopefully help illustrate the overall correlation in network metrics.

Point by point comments1. It should probably be mentioned somewhere in the introduction that social learning rules apply to either the social transmission of novel behavior (e.g., Aplin et al. 2015) or the social influence of others on behavior (e.g., Pike and Laland 2010, Danchin 2018), and that you aim specifically to look at social transmission.

We now ensure it is clear throughout our introduction that we consider the social transmission of ‘novel’ behaviour (e.g. lines 32, 112, 125).

2. The initial conditions of the simulations are not well enough explained before we get to the results. I was left wondering how the authors chose the first knowledgeable agent, which isn't answered until later.

In agreement with the editors, we re-structured our manuscript so that the methods section proceeds the Results section. Further, we substantially extended our methods section with a more detailed description and provide all equations and code (https://osf.io/6jrhz/) to replicate our results. We hope this improves the clarity of our methods.

3. The methods section could use more explanation. Those who are unfamiliar with NBDA would need to refer to other publications to see the equations, especially the meaning of 's=1', etc. Also, what are the other parameters set to (λ, A)? Consider including the equations, as well as a more thorough description of parameters. The same could be said for equations describing the network metrics.

We apologise for this shortcoming. We substantially extended our methods section with a more detailed description and provide all equations for the simulations (see section on ‘Simulations’) and code to replicate our results (https://osf.io/6jrhz/). Further, we included better definitions of all model parameters and also provide results on testing various parameters for the social transmission rate, ‘s’, frequency dependence parameter ‘f’, and threshold location ‘a’. In the main text, we present results on the parameters s=5 (social transmission rate), f=5 (frequency dependence parameter), a=5 (threshold location) and present results on smaller and larger parameters in the supplementary material (Figure S6-S15).

4. Related to point 4, what happens when A=0, in a pure social learning environment? This would reduce stochasticity due to asocial innovations, and would provide a pure test of the effect that authors predict arises from sociality and learning rules.

Please see our first response to the reviewer’s public comment:

“Finally, in regards to the point about asocial learning: ‘Asocial learning’ here is simply used to describe the acquisition of the behaviour by an individual that is not observed to take place due to the links within the observed social network. As such, some ‘asocial learning’ is necessary for the analysis to work at all, as without it there would be zero individuals with the behaviour to begin and this would persist throughout the simulation. Therefore, ‘allowing’ for asocial learning is a necessity, and then the question becomes how much asocial learning should be allowed for. We believe that the reviewer here is not arguing for not allowing asocial learning, but for only allowing for the very minimum amount of asocial learning (i.e. 1 asocial learning event per simulation per local population), and then banning asocial learning from that point onwards. However, we do not know of an animal system were such a scenario is likely to take place in natural settings, nor do we know of any previous empirical work that has dismissed the possibility of asocial learning entirely after the first asocial learning event. Therefore, we have opted to match what we believe would be relevant to the systems considered here, and continue allowing asocial learning after the first asocial learning event (indeed, this is also a major conceptual aspect of NBDA approaches generally too). Nevertheless, as we now provide all the data and code to replicate our simulations, the more interested readers are free to modify this to their desired asocial learning levels.”

5. L470 "foal" should be "focal".

Changed in line 229.

6. I actually think it's more helpful to show results when you standardize network size in the main text, and put Figure 2 in the supplement. Figure 2 is difficult to read, and something like Figure S2 is easier to interpret if you're assessing relative differences in diffusion dynamics. Also rather than presenting each network size as a color, select one network size (or a binned size) and present a variation metric (e.g., percentile intervals).

We agree that Figure S2 is easier to interpret for assessing the relative difference in diffusion dynamics. However, we were specifically interested in also presenting the effects of network size. Therefore, we decided to leave Figure 2 in the main text. However, we binned network size into different groups and now present the mean and the 95% confidence intervals as a measure of variation.

7. Suggest changing section title "Social network size and behavioral spreading" to match first section "Relationship between…". Also suggest "diffusion" rather than behavioural spreading. After reading the section, this seems more to do with how network size impacts the correlation between variables, rather than the diffusion itself. Maybe change the heading to reflect this?

We changed the heading to ‘Relationship between social network size and pathways of behavioural diffusion’ in line 433.

8. L204 – 231: Overall I think this section is fairly dense compared to the first section, and after reading it several times, I'm still not sure what I should take away from it. It looks like you have a very low N at large network sizes, which could drive some of these correlations in Figure 4. The fact that agents can asocially learn also makes it hard to interpret what these correlations mean.

We restructured this section to improve clarity in lines 433-457. Removing large networks (>=50 individuals) did not change the correlation coefficients substantially (see Figure 3).

9. L204-216: I had to read the beginning of this section several times, and it's more confusing than the first section of results. The results communicated until L210 do not relate to network size, and seem to repeat the previous section. I suggest removing this or incorporating it into the previous section and starting with how network size affected simulation dynamics.a. Also I suggest rewriting to avoid putting the variable of interest in parentheses (e.g., L 209, 213).

We restructured this section (375-457) and made changes to the variables in parentheses as suggested (lines 435-438).

b. L208: "the mean average network metric" was confusing -- do you mean "average network measure"?

We removed this sentence in the revised manuscript.

c. L210: I suggest "The direction and magnitude of the correlation between ind. Sociality and order of acquisition were predicted by network size. This relationship was modulated by transmission rule…" to improve clarity.

Changed as suggested in line 434-435.

10. Figure 4: I find it very hard to see all 4 lines, maybe choose different colors?

We changed the colours in the revised Figure 4 and present each social learning rule in a separate plot to improve visibility.

11. L224: Which means network metric?

We are not sure what the reviewer refers here to. We would be grateful if the reviewer could clarify.

12. L248: This information about initial conditions should come before the results.

This information was moved to the methods section which in accordance with the editors proceeds now the Results section (lines 302-320).

13. L248: "Spreading simulation" -> diffusion simulation.

Changes to ‘simulation’ in line 302.

14. L253: Without the equations written out in the methods, it's difficult to assess how the learning model works. Is asocial learning turned off under obligate social learning? It's my understanding that in NBDA, the s parameter controls the relative strength of social learning per unit connection to asocial learning. In the usual formalization of NBDA, the probability of asocial learning is constant in all individuals, contra L251 which states that asocial learning only occurs in unconnected individuals. Does your model assume that individuals who are "well-connected" (also undefined in the manuscript) have the $A$ parameter set to zero? If this is the case, the authors should include a justification/definition of being well-connected.

We apologise for the unclarity. We now include all relevant equations and rewrote several sections of the text in line with the other reviewers’ suggestions accordingly (see revised methods section). The statement ‘asocial learning only occurs in unconnected individuals’ is not true, but it is true that ‘only asocial learning occurs in unconnected individuals’, which is a big difference. We have ensured this is clear throughout. All individuals do have the same asocial learning rate and what we measure is at each event a new individual adopts the behaviour, what is the probability of this behavioural adoption steaming from social learning under each given social learning rule. If an individual has no social connections to any informed individuals, then its’ probability to adopt the behaviour via social learning is 0. Since in our simulations at every timestep a new individual learns, in such a case, the individual would adopt the behaviour via asocial learning.

15. L317: I'm still finding it hard to wrap my head around how a bird with low clustering is central and highly social. A nice way to explain/justify the differences between more and less social individuals would be to make a figure of an exemplar network, with several stereotypes highlighted, along with their social metrics.

We apologise for this unclarity. As mentioned above, we agree that classifying individuals with lower weighted clustering coefficients as less social is not clear. We re-wrote across the revised manuscript. Overall, weighted degree and weighted betweenness are positively correlated, and weighted degree and weighted clustering, and weighted betweenness and weighted clustering are negatively correlated in our social networks (Table S1). In addition, we added a supplementary figure (Figure S1) illustrating a few social network examples colour-coding the range of network metrics which will hopefully help illustrate the overall correlation in network metrics.

16. L332: Cantor et al. (2021, Proc. R. Soc. B) should probably also be cited here, as they measure the performance of recombination and subsequent diffusion.

Cited in line 585.

17. L342: Overall, this manuscript has synergy with the study "Cultural diffusion dynamics depend on behavioural production rules" (doi.org/10.1098/rspb.2022.1001), which explicitly explores the difference between acquisition and usage, and also uses NBDA as a generative model. It would be relevant to cite here.

Thank you for forwarding this relevant paper. We cited it across the revised manuscript (e.g. line 698).

18. Figure 5: If individuals have a low probability of social learning, do they have a high probability of asocial learning? Or not learning at all? Are there cases when both the probability of individual learning and social learning are low? Also, this is another case where normalizing the x axis between network sizes would be more informative. The authors might set asocial learning to 0 and simply directly measure the probability of acquisition by each naive agent at each time-step, since the manuscript is focused on social transmission rather than social transmission and asocial learning.

In our simulations, at each timestep a new individual adopts the behaviour. All individuals do have the same asocial learning rate and what we measure is at each event a new individual adopts the behaviour, what is the probability of this behavioural adoption steaming from social learning under each given social learning rule. If an individual has no social connections to any informed individuals, then its’ probability to adopt the behaviour via social learning is 0. Since in our simulations at every timestep a new individual learns, in such a case, the individual would adopt the behaviour via asocial learning.

19. Related to the interpretation of the model, the authors use the word "adopt" throughout the manuscript, although one could argue that their model is not of adoption, but of knowledge transmission, since there is no mechanism to determine whether individuals would actually use the behavior once acquiring knowledge of it. In other places, the authors have used the language of knowledge transmission (e.g., Figure 1 caption). It might be best to stick with knowledge transmission throughout the paper.

We agree that we cannot philosophically distinguish between knowledge transmission and behavioural adoption. The vast majority of empirical research on animal social learning probably refers to behavioural adoption because knowledge acquisition is difficult to measure in animals, whilst the adoption of a behaviour is observable. Many of our more complex learning rules refer to behavioural adoption rather than knowledge acquisition. For instance, under conformity learning, we expect an individual to only adopt a novel behaviour once the behaviour is performed by the majority of its’ social connections. In such a case, an individual may have already acquired the knowledge to perform a novel behaviour but will only adopt the behaviour once the majority of its’ social connections performs the behaviour. Further, it is the behavioural adoption (rather than the knowledge acquisition) which is the important part for transmitting the behaviour further along the network. Therefore, we decided to stick with behavioural adoption rather than knowledge transmission and re-wrote across the revised manuscript.